# Learning Latent Seasonal-Trend Representations for Time Series Forecasting

**Zhiyuan Wang**[1], **Xovee Xu**[1], **Weifeng Zhang**[1], **Goce Trajcevski**[2], **Ting Zhong**[1], **Fan Zhou**[1]*

[1] University of Electronic Science and Technology of China
[2] Iowa State University
zhy.wangcs@gmail.com, xovee@live.com, weifzh@outlook.com,
gocet25@iastate.edu, {tingzhong, fan.zhou}@uestc.edu.cn

## Abstract

Forecasting complex time series is ubiquitous and vital in a range of applications but challenging. Recent advances endeavor to achieve progress by incorporating various deep learning techniques (e.g., RNN and Transformer) into sequential models. However, clear patterns are still hard to extract since time series are often composed of several intricately entangled components. Motivated by the success of disentangled variational autoencoder in computer vision and classical time series decomposition, we plan to infer a couple of representations that depict seasonal and trend components of time series. To achieve this goal, we propose LaST, which, based on variational inference, aims to disentangle the seasonal-trend representations in the latent space. Furthermore, LaST supervises and disassociates representations from the perspectives of themselves and input reconstruction, and introduces a series of auxiliary objectives. Extensive experiments prove that LaST achieves state-of-the-art performance on time series forecasting task against the most advanced representation learning and end-to-end forecasting models. For reproducibility, our implementation is publicly available on Github[1].

## 1 Introduction

Time series forecasting plays a significant role in plethora of modern applications, ranging from climate analysis [1], energy production [2], traffic flows [3] to financial markets and various industrial systems [4]. The ubiquity and importance of time series data have recently attracted researcher efforts resulting in a myriad of deep learning forecasting models [5, 6] ameliorating the time series forecasting. Based on advanced techniques such as RNN and Transformer [7]), these methods usually learn latent representations to epitomize every instant of the signals, and then derive forecasting results by a predictor, achieving great progress on forecasting tasks.

However, these models have difficulties to extract exact/clear information related to temporal patterns (e.g., seasonality, trend, and level), especially in supervised end-to-end architecture without any constraint on representations [8]. As a consequence, efforts have been made to apply the variational inference into time series modeling [9, 10, 11], where improved guidance for latent representations with probabilistic form has been proved beneficial to downstream time series tasks [12]. However, when various intricately co-evolving constituents exist a time series data, analyzing with a single representation will result in superficial variable and models' non-reusability and lack of interpretability, due to the highly entangled nature of neural networks [13, 14]. Thus, while providing efficiency and effectiveness, existing approaches with a single high-dimensional representation sacrifice the

---

*Corresponding author.
[1] https://github.com/zhycs/LaST

information utilization and explainability, which may further lead to overfitting and degenerated performance.

To address the above limitations and seek a new disentangled time series learning framework, we leverage the ideas from the decomposition strategy [15, 16] to split time series data into several components, each of which captures an underlying pattern category. The decomposition assists the analysis process and reveals underlying insights more consistently with human intuition. This insight motivates us to produce a couple of latent representations that respond to different time series characteristics (in our case the seasonal and trend), from which we predict the results by formulating sequence as the sum of these characteristics. The representations should be as independent as possible to avoid a model prone to feature entanglement, while also having sufficient information to the input sequence.

Towards that, we propose a novel framework **LaST** to learn the **La**tent **S**easonal-**T**rend representations for time series forecasting. LaST exploits an encoder-decoder architecture and follows variational inference theory [17] to learn a couple of disentangled latent representations that describe seasonal and trend of time series. To achieve this goal, LaST enforces the representations into disentanglement under two constraints: (1) from the input reconstruction, we dissect intrinsic seasonal-trend patterns which can be readily obtained from raw time series and off-the-shelf measurement methods, and accordingly design a series of auxiliary objectives; (2) from representations themselves, we minimize the mutual information (MI) between seasonal and trend representations on the premise that grantee the consistency between input data and each of them. Our main contributions are threefold:

- We start with the variational inference and information theory to design the seasonal-trend representations learning and disentanglement mechanisms, and practically demonstrate their effectiveness and superiority (over the existing baselines) on time series forecasting task.

- We propose LaST, a novel latent seasonal-trend representations learning framework, which encodes input as disentangled seasonal-trend representations and provides a practicable approach that reconstructs seasonal and trend separately to avoid chaos.

- We introduce MI terms as a penalty and present a novel tractable lower bound and an upper bound for their optimizations. The lower bound ameliorates the biased gradient issue in prevalent MINE approach and ensures informative representations. The upper bound provides the feasibility to further reduce the overlapping of seasonal-trend representations.

## 2 Related work

Most of the deep learning methods for time series forecasting are designed as an end-to-end architecture. Various basic techniques (e.g., residual structure [18, 19], autoregressive network [20, 21], and convolutions [22, 23]) are exploited to produce expressive non-linear hidden states and embeddings that reflect the temporal dependencies and patterns. There is also a body of works that apply Transformer [7] structure into time series forecasting tasks [24, 25, 26, 6], aiming to discover the relationships across the sequence and focus on the important time points. Deep learning methods have achieved superior performance in comparison to the classical algorithms such as ARIMA [27] and VAR [28], and have become prevalent in multiple applications.

Learning flexible representations has been demonstrated to be beneficial for downstream tasks by numerous researches [12, 29]. In time series representations domain, early methods, employing the variational inference, jointly train an encoder and corresponding decoder that reconstructs raw signals to learn approximate latent representations [10, 30, 31]. Recent efforts have improve these variational methods [32, 33] by establishing more complex and flexible distributions using techniques such as copula [32] and normalizing flow [34, 35]. Another group of works exploited the burgeoning contrastive learning to obtain invariant representations from augmented time series [36, 37, 38], which avoids the reconstruction process and improves representations without additional supervision.

Time series decomposition [15, 16] is a classical method that splits complex time series into several components to obtain temporal patterns and interpretability. Recent works have applied machine learning and deep learning approaches [39, 40, 41] to robustly and efficiently achieve the decomposition on large-scale datasets. There are also research results that tackle forecasting with the assistance of decomposition. For example, Autoformer [26] decomposes time series into seasonal-trend parts by average pooling and introduces an autocorrelation mechanism to empower Transformer [7] for better

relations discovery. CoST [38] encodes signals into seasonal and trend representations in frequency and temporal domains, respectively, and introduces contrastive learning to supervise their learning. Different from our work, these methods exploit simple average pooling decomposition mechanism which may provide incompatible periodical assumptions, or intuitively disentangle the representations by processing in different domains. Meanwhile, LaST adaptively epitomizes the seasonality and trend by disentangled representations and boosts their disassociation from a probabilistic perspective in the latent space.

## 3 Latent seasonal-trend representations learning framework

We now formalize the problem definition and introduce the proposed LaST framework. We note that in LaST we use seasonal and trend characteristics for disentanglement learning but our framework can be easily extended to adapt to situations that have more than two components to dissociate.

**Problem definition.** Consider a time series dataset $\mathcal{D}$ consisting of $N$ i.i.d. sequences denoted as $X_{1:T}^{(i)} = \{x_1^{(i)}, x_2^{(i)}, \cdots, x_t^{(i)}, \cdots, x_T^{(i)}\}$, where $i \in \{1, 2, \ldots, N\}$, and each $x_t^{(i)}$ is univariate or multivariate value representing the current observation(s) at time instant $t$ (e.g., price and rainfall). We aim to derive a model that outputs expressive representations $Z_{1:T}$, suitable for predicting future sequence(s) $Y = \hat{X}_{T+1:T+\tau}$. Hereafter, when there is no ambiguity we omit the superscripts and the subscript $_{1:T}$. A model that infers the likelihood between observation $X$ and future $Y$ with latent representation $Z$ can be formulated as follows:

$$P(X, Y) = P(Y|X)P(X) = \int_Z P(Y|Z)P(Z|X)dZ \int_Z P(X|Z)P(Z)dZ. \tag{1}$$

From the perspective of variational inference (cf. [17]), the likelihood $P(X|Z)$ is calculated by a posterior distribution $Q_\phi(Z|X)$ and maximized by the following evidence lower bound (ELBO):

$$\log P_\Theta(X, Y) \geq \log \int_Z P_\psi(Y|Z)Q_\phi(Z|X)dZ + \mathbb{E}_{Q_\phi(Z|X)}[\log P_\theta(X|Z)]$$
$$- \mathbb{KL}(Q_\phi(Z|X)||P(Z)) = \mathcal{L}_{ELBO}, \tag{2}$$

where $\Theta$ is composed of $\psi$, $\phi$, and $\theta$ denotes learned parameters.

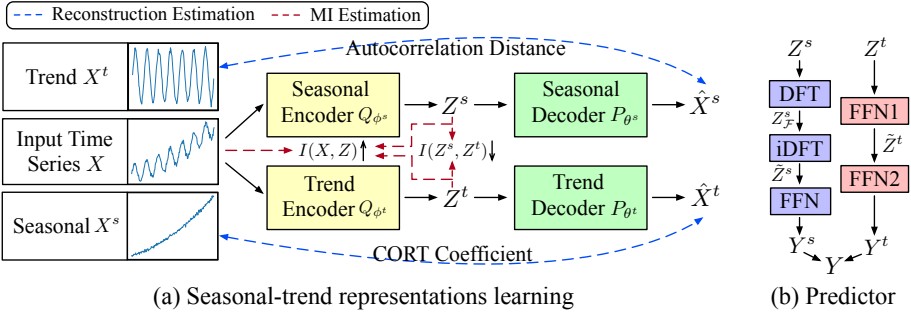

(a) Seasonal-trend representations learning       (b) Predictor

Figure 1: Overview of proposed LaST framework.

However, as pointed out in Sec. 1, this faces an entanglement problem and cannot clearly extract complicated temporal patterns. To ameliorate this limitation, we incorporate the decomposition strategy into our LaST that learns a couple of disentangled representations to depict seasonal and trend dynamics. Specifically, we formulate the temporal signals $X$ and $Y$ as the sum of seasonal and trend components, i.e., $X = X^s + X^t$. Accordingly, the latent representation $Z$ is factorized into $Z^s$ and $Z^t$, assumed to be independent of each other – i.e., $P(Z) = P(Z^s)P(Z^t)$. Figure 1 illustrates the two parts of the LaST framework: (a) representations learning, producing disentangles seasonal-trend representations (separate reconstructions and MI constraints); and (b) prediction based on learned representations.

**Theorem 1.** *With the decomposition strategy, Eq.* (2) *(i.e., the ELBO) naturally has the following factorized form:*

$$\mathcal{L}_{ELBO} = \log \int_{Z^s} \int_{Z^t} P_\psi(Y|Z^s, Z^t) Q_{\phi^s, \phi^t}(Z^s, Z^t|X) dZ^s dZ^t \qquad \text{(predictor)} \quad (3)$$

$$+ \mathbb{E}_{Q_{\phi^s}(Z^s|X)}[\log P_{\theta^s}(X^s|Z^s)] + \mathbb{E}_{Q_{\phi^t}(Z^t|X)}[\log P_{\theta^t}(X^t|Z^t)] \quad \text{(reconstruction)} \quad (4)$$

$$- \mathbb{KL}(Q_{\phi^s}(Z^s|X)||P(Z^s)) - \mathbb{KL}(Q_{\phi^t}(Z^t|X)||P(Z^t)). \qquad \text{(KL divergence)} \quad (5)$$

The detailed inference process of the above formula is provided in Appendix A.1. The ELBO is split into three main units, i.e., Eqs. (3), (4), and (5). The *predictor* makes forecasting and measures the accuracy (e.g,. L1 or L2 losses), *reconstruction* and *KL divergence* are served as regularization terms aiming to improve the learned representations. The three units are described in the following.

**Predictor:** The *predictor* (cf. Eq. (3) and Figure 1(b)) can be regarded as the sum of two independent parts: $\log \int_{Z^s} P_{\psi^s}(Y^s|Z^s) Q_{\phi^s}(Z^s|X) dZ^s$ and $\log \int_{Z^t} P_{\psi^t}(Y^t|Z^t) Q_{\phi^t}(Z^t|X) dZ^t$. Here we introduce two specialized approaches to harness the seasonal-trend representations combining their own characteristics. Given the seasonal latent representation $Z^s \in \mathbb{R}^{T \times d}$, the seasonal predictor first employs the discrete Fourier transform (DFT) algorithm to detect the seasonal frequencies, i.e., $Z_{\mathcal{F}}^s = \text{DFT}(Z^s) \in \mathbb{C}^{F \times d}$, where $F = \lfloor \frac{T+1}{2} \rfloor$ due to the Nyquist theorem [42]. Then, we inverse the frequencies back to the temporal domain to extend the representation to the future part, i.e., $\tilde{Z}^s = \text{iDFT}(Z_{\mathcal{F}}^s) \in \mathbb{R}^{\tau \times d}$. More details of the DFT and iDFT functions can be found in Appendix B.1. Given $Z^t$, trend predictor provides a feed forward network (FFN) $f : T \to \tau$ to produce a predictable representation $\tilde{Z}^t \in \mathbb{R}^{\tau \times d}$. We end the predictor with two FFNs to map $\tilde{Z}^s$ and $\tilde{Z}^t$ into $Y^s$ and $Y^t$, respectively, and obtain the forecasting result $Y$ by their sum.

**Reconstruction and KL divergence:** Among these two terms, the *KL divergence* can be easily estimated by Monte Carlo sampling with prior assumptions. Here we take a widely used setting that priors both follow $\mathcal{N}(0, I)$ for efficiency, more discussions of our priors can be found in Appendix C. As for *reconstruction* term, it cannot be directly measured owing to the unknown $X^s$ and $X^t$. Besides, merging these two terms into $\mathbb{E}_{Q_{\phi^s, \phi^t}(Z^s, Z^t|X)}[\log P_{\theta^s, \theta^t}(X|Z^s, Z^t)]$ will result in chaos since the decoder is prone to reconstruct the intricate time series from every representation.

**Theorem 2.** *With the Gaussian distribution assumption, the reconstruction loss $\mathcal{L}_{rec}$ can be estimated without leveraging $X^s$ and $X^t$, and Eq.* (4) *can be replaced with the following formula:*

$$\mathcal{L}_{rec} = -\sum_{\kappa=1}^{T-1} \left\| \mathcal{A}_{XX}(\kappa) - \mathcal{A}_{\hat{X}^s \hat{X}^s}(\kappa) \right\|^2 + CORT(X, \hat{X}^t) - \left\| \hat{X}^t + \hat{X}^s - X \right\|^2, \qquad (6)$$

$$CORT(X, \hat{X}^t) = \frac{\sum_{i=1}^{T-1} \Delta X_i^t \Delta \hat{X}_i^t}{\sqrt{\sum_{i=1}^{T-1} \Delta X^t} \sqrt{\sum_{i=1}^{T-1} \Delta \hat{X}^t}}, \qquad (7)$$

*where $\mathcal{A}_{XX}(\kappa) = \sum_{i=1}^{T-\kappa}(X_t - \bar{X})(X_{t+\kappa} - \bar{X})$ is the autocorrelation coefficient with lagged value $\kappa$ (we employ an efficient implementation in frequency domain [43], details are in Appendix B.2), $CORT(X, \hat{X}^t)$ is the temporal correlation coefficient, and $\Delta X_i = X_i - X_{i-1}$ is the first difference.*

The proof is provided in Appendix A.2. According to Eq. (6), the reconstruction loss now can be estimated and, conversely, used to supervise disentangled representation learning. However, we find that this framework still holds certain drawbacks: (1) The KL divergence tends to narrow down the distance between posterior and prior. The modeling choice tends to sacrifice the variational inference vs. data fit when modeling capacity is not sufficient to achieve both [44]. The posterior may become almost non-informative for the inputs, which causes the forecastings irrelevant to the observations. (2) The disentanglement of the seasonal-trend representations is boosted indirectly by the separate reconstruction, where we need to impose a direct constraint on the representations themselves. We alleviate these limitations by introducing additional mutual information regularization terms. Specifically, we increase the mutual information between $Z^s$, $Z^t$ and $X$ to alleviate the divergence narrowing problem [44, 45], while decreasing mutual information between $Z^s$ and $Z^t$ to further dissociate their representations. The maximizing objective of LaST becomes

$$\mathcal{L}_{LaST} = \mathcal{L}_{ELBO} + I(X, Z^s) + I(X, Z^t) - I(Z^s, Z^t), \qquad (8)$$

where $I(\cdot, \cdot)$ denotes the mutual information between two representations. However, the two mutual information terms are untraceable [46, 47, 48]. We address this problem in the next section.

# 4 Mutual information bounds for optimization

We now address the traceable mutual information bounds, maximizing $I(X, Z^s)$ and $I(X, Z^t)$, and minimizing $I(Z^s, Z^t)$ in Eq. (8), and provide lower and upper bounds for the model optimization.

**Lower bound for $I(X, Z^s)$ or $I(X, Z^t)$.** We omit the superscript $s$ or $t$ when analyzing lower bound. Among the prior approaches exploring the lower bounds for MI [49, 50, 51], MINE [51], for example, employs KL divergence between the joint distribution and marginals and defines an energy-based variational family to achieve a flexible and scalable lower bound. This can be formulated as $I(X, Z) \geq \mathbb{E}_{Q_\phi(X,Z)}[\gamma_\alpha(X, Z)] - \log \mathbb{E}_{Q(x)Q_\phi(z)}[e^{\gamma_\alpha(X,Z)}] = I_{\text{MINE}}$ , where $\gamma_\alpha$ is a learned normalized critic with parameters $\alpha$. However, this bound suffers from the biased gradient owing to the parametric logarithmic term (see Appendix A.3 for proof). Inspired by [47], we substitute the logarithmic function by its tangent family to ameliorate the above biased bound:

$$I_{\text{MINE}} \geq \mathbb{E}_{Q_\phi(X,Z)}[\gamma_\alpha(X, Z)] - (\frac{1}{\eta}\mathbb{E}_{Q(x)Q_\phi(z)}[e^{\gamma_\alpha(X,Z)}] + \log \eta - 1)$$

$$\geq \mathbb{E}_{Q_\phi(X,Z)}[\gamma_\alpha(X, Z)] - \frac{1}{\eta}\mathbb{E}_{Q(x)Q_\phi(z)}[e^{\gamma_\alpha(X,Z)}], \tag{9}$$

where $\eta$ denotes the different tangent points. The first inequality relies on the concave negative logarithmic function – the values on the curve are upper bounds for that on the tangent line, and is tight when the tangent point overlaps the independent variable, i.e., the true value of $\mathbb{E}_{Q(x)Q(z)}[e^{\gamma(X,Z)}]$. The closer the distance between tangent point and independent variable, the greater the lower bound. Therefore, we set $\eta$ as the variational term $\mathbb{E}_{Q(x)Q_\phi(z)}[e^{\gamma_\alpha(X,Z)}]$ that estimates the independent variable to obtain as great lower bound as possible. In the second inequality, $\gamma_\alpha(x, z)$ – a critic function activated by Sigmoid – is limited within $[0, 1]$ and thus $-(\log \eta - 1) \geq 0$. This inequality is tight only if $\mathbb{E}_{Q(x)Q_\phi(z)}[\gamma_\alpha(X, Z)] = 1$, which means $\gamma_\alpha$ can discriminate whether a pair of variables $(X, Z)$ is sampled from the joint distribution or marginals. Similarly to MINE, this consistency problem can be addressed by the universal approximation theorem for neural networks [52]. Thus, Eq. (9) provides a flexible and scalable lower bound for $I(X, Z)$ with an unbiased gradient.

For the evaluation, we exploit a traceable manner [53, 51] that draws joint samples $(X^{(i)}, Z^{(i)})$ by $Q(Z^{(i)}|X^{(i)})P_\mathcal{D}(X^{(i)})$. As for the marginal $Q_\phi(Z)$, we randomly select a datapoint $j$ and then sample it from $Q_\phi(Z|X^{(j)})P_\mathcal{D}(X^{(j)})$. Details of the optimization are shown in Algorithm 1.

**Upper bound for $I(Z^s, Z^t)$.** Few efforts have been made that explore the traceable upper bound for mutual information [54, 47, 55]. Existing upper bounds (listed in Appendix D.1) are traceable with known probabilistic density of joint or conditional distributions here being $Q(Z^s|Z^t)$, $Q(Z^t|Z^s)$ or $Q(Z^s, Z^t)$. However, these distributions lack interpretability and can hardly be directly modeled, which leads to untraceable estimations of the above upper bounds.

To avoid the direct estimation of unknown probabilistic densities, we introduce an energy-based variational family for $Q(Z^s, Z^t)$ that uses a normalized critic $\gamma_\beta(Z^s, Z^t)$ like Eq. (9) to establish a traceable upper bound. Specifically, we incorporate the critic $\gamma_\beta$ into the upper bound $I_{\text{CLUB}}$ [55] to obtain a traceable Seasonal-Trend Upper Bound (STUB) for $I(Z^s, Z^t)$, which is defined as:

$$I(Z^s, Z^t) \leq \mathbb{E}_{Q(Z^s,Z^t)}[\log Q(Z^s|Z^t)] - \mathbb{E}_{Q(Z^s)Q(Z^t)}[\log Q(Z^s|Z^t)] = I_{\text{CLUB}} \tag{10}$$

$$= \mathbb{E}_{Q_{\phi^s,\phi^t}(Z^s,Z^t)}[\gamma_\beta(Z^s, Z^t)] - \mathbb{E}_{Q_{\phi^s}(Z^s)Q_{\phi^t}(Z^t)}[\gamma_\beta(Z^s, Z^t)] = I_{\text{STUB}}. \tag{11}$$

The derivation details of this formula are provided in Appendix D.2. The inequality in Eq. (10) is tight only if $Z^s$ and $Z^t$ are a pair of independent variables [55]. This is exactly a sufficient condition for $I_{\text{STUB}}$, since MI and Eq. (11) are both zeros on the independent situation, which is our seasonal-trend disentanglement optimal objective. The critic $\gamma_\beta$, similar to $\gamma_\alpha$, takes on the discriminating responsibility but provides converse scores, constraining the MI to a minimum. However, Eq. (11) may get negative values during the learning of parameter $\beta$, resulting an invalid upper bound for MI. To alleviate this problem, we additionally introduce a penalty term $\|I_{STUB}^{neg}\|^2$ to assist the model optimization, which is an L2 loss of the negative parts in $I_{\text{STUB}}$.

For the evaluation, we take the same sampling manner as the one in the lower bound and optimization details are also shown in Algorithm 1.

---

**Algorithm 1** An epoch of the optimization of LaST.

---

1: Initialize the parameters of LaST: $\Theta = \{\psi^s, \psi^t, \phi^s, \phi^t, \theta^s, \theta^t\}, \Gamma = \{\alpha^s, \alpha^t, \beta\}$.
2: **for** a mini-batch with size $\mathcal{B}$ consisting of $\{X^{(i)}, Y^{(i)}\}_{i \in \mathcal{B}}$ **in** training set **do**
3:      Get samples of the latent representations $\{Z^{s(i)}\}_{i \in \mathcal{B}}$ and $\{Z^{t(i)}\}_{i \in \mathcal{B}}$ from distributions $\{Q_{\phi^s}(Z^s|X^{(i)})\}_{i \in \mathcal{B}}$ and $\{Q_{\phi^t}(Z^t|X^{(i)})\}_{i \in \mathcal{B}}$, respectively;
4:      Shuffle the $\{Z^{s(i)}\}_{i \in \mathcal{B}}$ and $\{Z^{t(i)}\}_{i \in \mathcal{B}}$ and form $\{Z^{s(j)}\}_{j \in \mathcal{B}}$ and $\{Z^{t(j)}\}_{j \in \mathcal{B}}$, respectively;
5:      Compute the $\eta^s, \eta^t$: $\eta^s \leftarrow \frac{1}{\mathcal{B}} \sum_{i=j=1}^{\mathcal{B}} e^{\gamma_{\alpha^s}(X^{(i)}, Z^{s(j)})}, \eta^t \leftarrow \frac{1}{\mathcal{B}} \sum_{i=j=1}^{\mathcal{B}} e^{\gamma_{\alpha^t}(X^{(i)}, Z^{t(j)})}$;
6:      Update parameters $\Theta$: $\Theta \leftarrow G(\nabla_\Theta)[\mathcal{L}_{ELBO} + \frac{1}{\mathcal{B}} \sum_{i=j=1}^{\mathcal{B}} (\gamma_{\alpha^s}(X^{(i)}, Z^{s(i)}) - \frac{1}{\eta} e^{\gamma_{\alpha^s}(X^{(i)}, Z^{s(j)})} + \gamma_{\alpha^t}(X^{(i)}, Z^{t(i)}) - \frac{1}{\eta} e^{\gamma_{\alpha^t}(X^{(i)}, Z^{t(j)})}) - \frac{1}{\mathcal{B}} \sum_{i=j=1}^{\mathcal{B}} (\gamma_\beta(Z^{s(i)}, Z^{t(i)}) - \gamma_\beta(Z^{s(i)}, Z^{t(j)})) + average(\|(\gamma_\beta(Z^{s(i)}, Z^{t(i)}) - \gamma_\beta(Z^{s(i)}, Z^{t(j)}))^{neg}\|^2)]$;
7:      Update parameters $\Gamma$: $\Gamma \leftarrow G(\nabla_\Gamma)[\frac{1}{\mathcal{B}} \sum_{i=j=1}^{\mathcal{B}} (\gamma_{\alpha^s}(X^{(i)}, Z^{s(i)}) - \frac{1}{\eta} e^{\gamma_{\alpha^s}(X^{(i)}, Z^{s(j)})} + \gamma_{\alpha^t}(X^{(i)}, Z^{t(i)}) - \frac{1}{\eta} e^{\gamma_{\alpha^t}(X^{(i)}, Z^{t(j)})}) - \frac{1}{\mathcal{B}} \sum_{i=j=1}^{\mathcal{B}} (\gamma_\beta(Z^{s(i)}, Z^{t(i)}) - \gamma_\beta(Z^{s(i)}, Z^{t(j)}) + average(\|(\gamma_\beta(Z^{s(i)}, Z^{t(i)}) - \gamma_\beta(Z^{s(i)}, Z^{t(j)}))^{neg}\|^2)]$;
8: **end for**

---

# 5 Experiments

We now present the results of our extensive experimental evaluations comparing LaST with state-of-the-art baselines and report a series of empirical results, along with ablation study and visualizations of seasonal-trend representations. Further details and results are provided in Appendix F.

## 5.1 Settings

**Datasets and Baselines.** We conducted our experiments on seven real-world benchmark datasets from four categories of mainstream time series forecasting applications: (1) **ETT** [2][25]: Electricity Transformer Temperature consists of the target value "oil temperature" and six "power load" features, recorded hourly (i.e., ETTh1 and ETTh2) and every 15 minutes (i.e., ETTm1 and ETTm2) over two years. (2) **Electricity**, from the UCI Machine Learning Repository [3] and preprocessed by [56], is composed of the hourly electricity consumption of 321 clients in kWh from 2012 to 2014. (3) **Exchange** [56] with daily exchange rates of eight countries from 1990 to 2016. (4) **Weather** [4] contains 21 meteorological indicators (e.g., temperature and humidity) and is recorded every 10 minutes in 2020. We compare our LaST with the latest state-of-the-art methods on time series modeling and forecasting tasks from two categories: (1) representation learning techniques, including COST [38], TS2Vec [37], and TNC [36]; (2) end-to-end forecasting models, including VAE-GRU [10], Autoformer [26], Informer [25], and TCN [22]. Further descriptions and settings of these baselines are provided in appendix F.1.

**Evaluation setup.** Following the prior work, we run our model on both univariate and multivariate forecasting settings. In multivariate forecasting, LaST accepts and forecasts all variables in datasets. In univariate forecasting, LaST only considers a specific feature in each dataset. We employ the standard normalization and set input length $T = 201$ for all datasets. For the dataset split, we follow a standard protocol that categorizes all datasets into training, validation, and test set in chronological order by the ratio of 6:2:2 for all datasets. We report the evaluation results on the test set while the model achieves the best performance on the validation set.

**Implementation details.** As for the network structure of LaST, we use a single-layer fully connected network as the feed forward network (FFN), which is applied in the modeling of posterior, reconstruction, and predictor. Besides, we employ the 2-layer MLP for the critic $\gamma$ in MI bound estimations. Dimensions of seasonal and trend representations are consistent. We set them as 32 in univariate forecasting and as 128 in multivariate forecasting on other datasets. MAE loss is used to measure the

---

[2] https://github.com/zhouhaoyi/ETDataset
[3] https://archive.ics.uci.edu/ml/datasets/ElectricityLoadDiagrams
[4] https://www.bgc-jena.mpg.de/wetter

Table 1: Univariate forecasting comparisons. Complete results on ETT benchmark are shown in appendix F.4. Best performance is highlighted in bold font and the second best results are underlined.

| Method | | LaST | | CoST | | TS2Vec | | TNC | | VAE-GRU | | Autoformer | | Informer | | TCN | |
|---|---|---|---|---|---|---|---|---|---|---|---|---|---|---|---|---|---|
| | | MSE | MAE | MSE | MAE | MSE | MAE | MSE | MAE | MSE | MAE | MSE | MAE | MSE | MAE | MSE | MAE |
| ETTh1 | 24 | **0.030** | **0.131** | 0.040 | 0.152 | 0.039 | 0.151 | 0.057 | 0.184 | 0.042 | 0.155 | 0.057 | 0.189 | 0.098 | 0.247 | 0.104 | 0.254 |
| | 48 | **0.051** | **0.169** | 0.060 | 0.186 | 0.062 | 0.189 | 0.094 | 0.239 | 0.077 | 0.218 | 0.070 | 0.207 | 0.158 | 0.319 | 0.206 | 0.366 |
| | 168 | **0.078** | **0.211** | 0.097 | 0.236 | 0.142 | 0.291 | 0.171 | 0.329 | 0.172 | 0.344 | 0.108 | 0.260 | 0.183 | 0.346 | 0.462 | 0.586 |
| | 336 | **0.100** | **0.246** | 0.112 | 0.258 | 0.160 | 0.316 | 0.179 | 0.345 | 0.140 | 0.301 | 0.119 | 0.281 | 0.222 | 0.387 | 0.422 | 0.564 |
| | 720 | 0.138 | 0.298 | 0.148 | 0.306 | 0.179 | 0.345 | 0.235 | 0.408 | 0.204 | 0.381 | **0.109** | **0.264** | 0.269 | 0.435 | 0.438 | 0.578 |
| ETTm1 | 24 | **0.011** | **0.077** | 0.015 | 0.088 | 0.016 | 0.093 | 0.019 | 0.103 | 0.013 | 0.082 | 0.022 | 0.115 | 0.030 | 0.137 | 0.027 | 0.127 |
| | 48 | **0.021** | **0.108** | 0.025 | 0.117 | 0.028 | 0.126 | 0.045 | 0.162 | 0.026 | 0.120 | 0.032 | 0.138 | 0.069 | 0.203 | 0.040 | 0.154 |
| | 96 | **0.033** | **0.134** | 0.038 | 0.147 | 0.045 | 0.162 | 0.054 | 0.178 | 0.046 | 0.164 | 0.045 | 0.168 | 0.194 | 0.372 | 0.097 | 0.246 |
| | 288 | **0.069** | **0.197** | 0.077 | 0.209 | 0.095 | 0.235 | 0.142 | 0.290 | 0.127 | 0.294 | 0.071 | 0.207 | 0.401 | 0.554 | 0.305 | 0.455 |
| | 672 | **0.100** | **0.239** | 0.113 | 0.257 | 0.142 | 0.290 | 0.136 | 0.290 | 0.217 | 0.399 | 0.102 | 0.254 | 0.512 | 0.644 | 0.445 | 0.576 |
| Electricity | 24 | **0.151** | **0.277** | 0.243 | 0.264 | 0.260 | 0.288 | 0.252 | 0.278 | 0.330 | 0.406 | 0.290 | 0.411 | 0.251 | 0.275 | 0.243 | 0.367 |
| | 48 | **0.186** | **0.307** | 0.292 | 0.300 | 0.313 | 0.321 | 0.300 | 0.308 | 0.437 | 0.481 | 0.310 | 0.408 | 0.346 | 0.339 | 0.283 | 0.397 |
| | 168 | **0.243** | **0.346** | 0.405 | 0.375 | 0.429 | 0.392 | 0.412 | 0.384 | 0.433 | 0.476 | 0.435 | 0.490 | 0.544 | 0.424 | 0.357 | 0.449 |
| | 336 | **0.286** | **0.379** | 0.560 | 0.473 | 0.565 | 0.478 | 0.548 | 0.466 | 0.472 | 0.504 | 0.646 | 0.606 | 0.713 | 0.512 | 0.355 | 0.446 |
| | 720 | **0.322** | **0.422** | 0.889 | 0.645 | 0.863 | 0.651 | 0.859 | 0.651 | 0.543 | 0.563 | 0.609 | 0.587 | 1.182 | 0.806 | 0.387 | 0.477 |
| Average | | **0.121** | **0.236** | 0.208 | 0.268 | 0.222 | 0.289 | 0.234 | 0.328 | 0.219 | 0.326 | 0.201 | 0.306 | 0.345 | 0.400 | 0.278 | 0.403 |

forecasting derived from the predictor. For the training strategy, we use the Adam [57] optimizer, and training process is early stopped within 10 epochs. We initialize the learning rate with $10^{-3}$ and decay it with 0.95 weight every epoch.

## 5.2 Performance comparisons and model analysis

**Effectiveness.** Tables 1 and 2 summarize the results of univariate and multivariate forecastings respectively. LaST achieves state-of-the-art performance against the advanced representation baselines on five real-world datasets. The relative improvements on MSE and MAE are 25.6% and 22.1% against the best representation learning method CoST and are 22.0% and 18.9% against the best end-to-end models Autoformer. We note that Autoformer achieves better performance on long horizons forecasting on hourly ETT datasets and think there are two reasons: (1) Transformer-based models intrinsically establish long-range dependencies, which plays a crucial role in long sequence forecasting; (2) it employs a simple decomposition by average pooling with a fixed kernel size, which is more suitable for strongly periodic datasets like hourly ETT. This phenomenon is beneficial to long-term forecasting but limits the sensitivity to local context, and the bonus does not have significant impact on other datasets. Compared with baselines, LaST extracts the seasonal and trend patterns with disentangled representations adaptively and thus can be applied to intricate time series.

**Ablation study.** We investigated the performance benefits brought by each mechanism of LaST on a synthetic dataset (generation details are provided in appendix F.3) and ETTh1. The results are shown in Table 3, consisting of two groups: **M1** validates the mechanisms of seasonal-trend representations learning framework. In it, "w/o seasonal" and "w/o trend" denote LaST without the seasonal and trend components respectively; "w/o coe" denotes LaST without autocorrelation and CORT coefficients while estimating the reconstruction loss. **M2** judges the introduction and estimations of MI, where"w/o lower" and "w/o upper" indicate the removal of the lower and upper bounds for MI in regularization terms respectively; "with MINE" denotes that we replace our lower bound with MINE. The results show that all mechanisms improve the performance on the forecasting task. We notice that the quality drops a lot when removing the trend component. The reason is that seasonal forecasting derives from the iDFT algorithm, which is essentially a periodical repetition of historical observations. However, it captures the seasonal patterns and assists the trend component in complete LaST to bring the superiority, especially in the long-term settings and strongly periodical synthetic dataset. Besides, we observe that with biased regularization term MINE, the performance becomes unstable and sometimes even worse than LaST without MI lower bound, while our unbiased bound (cf. Eq.(9)) continuously outperforms it.

**Representation disentanglement.** We visualize the seasonal-trend representations with the t-SNE [58] technique in Figure 2. We also visualize the embeddings in last layer of Autoformer decoder as a comparison. The points with same color have a clearer and closer clustering in LaST, while they mix together without decomposition mechanisms ("w/o dec" indicates removal of the two decomposition mechanisms (autocorrelation and CORT coefficients, and the upper bound to MI).

Table 2: Multivariate forecasting comparisons. Complete results on ETT benchmark are shown in appendix F.4. Best performance is highlighted in bold font and the second best results are underlined.

| Method | **LaST** MSE | **LaST** MAE | CoST MSE | CoST MAE | TS2Vec MSE | TS2Vec MAE | TNC MSE | TNC MAE | VAE-GRU MSE | VAE-GRU MAE | Autoformer MSE | Autoformer MAE | Informer MSE | Informer MAE | TCN MSE | TCN MAE |
|---|---|---|---|---|---|---|---|---|---|---|---|---|---|---|---|---|
| ETTh1 24 | **0.324** | **0.368** | 0.386 | 0.429 | 0.590 | 0.531 | 0.708 | 0.592 | 0.529 | 0.534 | 0.384 | 0.428 | 0.577 | 0.549 | 0.583 | 0.547 |
| 48 | **0.351** | **0.380** | 0.437 | 0.464 | 0.624 | 0.555 | 0.749 | 0.619 | 0.612 | 0.593 | 0.392 | 0.419 | 0.685 | 0.625 | 0.670 | 0.606 |
| 168 | **0.468** | **0.453** | 0.643 | 0.582 | 0.762 | 0.639 | 0.884 | 0.699 | 0.758 | 0.647 | 0.490 | 0.481 | 0.931 | 0.752 | 0.811 | 0.680 |
| 336 | 0.566 | 0.512 | 0.812 | 0.679 | 0.931 | 0.728 | 1.020 | 0.768 | 0.844 | 0.692 | **0.505** | **0.484** | 1.128 | 0.873 | 1.132 | 0.815 |
| 720 | 0.740 | 0.650 | 0.970 | 0.771 | 1.063 | 0.799 | 1.157 | 0.830 | 1.045 | 0.816 | **0.498** | **0.500** | 1.215 | 0.896 | 1.165 | 0.813 |
| ETTm1 24 | **0.218** | **0.289** | 0.246 | 0.329 | 0.453 | 0.444 | 0.522 | 0.472 | 0.509 | 0.452 | 0.383 | 0.403 | 0.453 | 0.444 | 0.522 | 0.472 |
| 48 | **0.280** | **0.329** | 0.331 | 0.386 | 0.592 | 0.521 | 0.695 | 0.567 | 0.642 | 0.543 | 0.454 | 0.453 | 0.494 | 0.503 | 0.542 | 0.508 |
| 96 | **0.323** | **0.360** | 0.378 | 0.419 | 0.635 | 0.554 | 0.731 | 0.595 | 0.600 | 0.540 | 0.481 | 0.463 | 0.678 | 0.614 | 0.666 | 0.578 |
| 288 | **0.392** | **0.403** | 0.472 | 0.486 | 0.693 | 0.597 | 0.818 | 0.649 | 0.769 | 0.678 | 0.634 | 0.528 | 1.056 | 0.786 | 0.991 | 0.735 |
| 672 | **0.491** | **0.466** | 0.620 | 0.574 | 0.782 | 0.653 | 0.932 | 0.712 | 0.799 | 0.673 | 0.606 | 0.542 | 1.192 | 0.926 | 1.032 | 0.756 |
| Electricity 24 | **0.125** | **0.222** | 0.136 | 0.242 | 0.287 | 0.375 | 0.354 | 0.423 | 0.190 | 0.250 | 0.165 | 0.286 | 0.312 | 0.387 | 0.235 | 0.346 |
| 48 | **0.146** | **0.245** | 0.153 | 0.258 | 0.309 | 0.391 | 0.376 | 0.438 | 0.228 | 0.280 | 0.178 | 0.295 | 0.392 | 0.431 | 0.253 | 0.359 |
| 168 | **0.170** | **0.265** | 0.175 | 0.275 | 0.335 | 0.410 | 0.402 | 0.456 | 0.240 | 0.297 | 0.215 | 0.327 | 0.515 | 0.509 | 0.278 | 0.372 |
| 336 | **0.188** | **0.280** | 0.196 | 0.296 | 0.351 | 0.422 | 0.417 | 0.466 | 0.262 | 0.318 | 0.218 | 0.329 | 0.759 | 0.625 | 0.287 | 0.382 |
| 720 | **0.223** | **0.309** | 0.232 | 0.327 | 0.378 | 0.440 | 0.442 | 0.483 | 0.296 | 0.347 | 0.252 | 0.356 | 0.969 | 0.788 | 0.287 | 0.381 |
| Exchange 24 | **0.033** | **0.122** | **0.033** | 0.127 | 0.108 | 0.252 | 0.105 | 0.236 | 0.064 | 0.178 | 0.060 | 0.178 | 0.611 | 0.626 | 2.483 | 1.327 |
| 48 | **0.056** | **0.162** | 0.058 | 0.165 | 0.200 | 0.341 | 0.162 | 0.270 | 0.133 | 0.262 | 0.091 | 0.222 | 0.680 | 0.644 | 2.328 | 1.256 |
| 168 | **0.190** | **0.320** | 0.198 | 0.327 | 0.412 | 0.492 | 0.397 | 0.480 | 0.334 | 0.432 | 0.405 | 0.473 | 1.097 | 0.825 | 2.372 | 1.279 |
| 336 | **0.430** | **0.482** | 0.512 | 0.523 | 1.339 | 0.901 | 1.008 | 0.866 | 0.614 | 0.606 | 0.509 | 0.524 | 1.672 | 1.036 | 3.113 | 1.459 |
| 720 | 1.521 | **0.898** | 1.855 | 0.998 | 2.114 | 1.125 | 1.989 | 1.063 | 2.285 | 1.117 | **1.447** | 0.941 | 2.478 | 1.310 | 3.150 | 1.458 |
| Weather 24 | **0.105** | **0.134** | 0.293 | 0.369 | 0.170 | 0.309 | 0.200 | 0.312 | 0.117 | 0.147 | 0.180 | 0.263 | 0.162 | 0.235 | 0.170 | 0.287 |
| 48 | **0.131** | **0.174** | 0.558 | 0.515 | 0.231 | 0.375 | 0.284 | 0.367 | 0.227 | 0.270 | 0.241 | 0.310 | 0.348 | 0.400 | 0.327 | 0.365 |
| 168 | **0.197** | **0.238** | 0.812 | 0.671 | 0.470 | 0.532 | 0.475 | 0.502 | 0.234 | 0.280 | 0.295 | 0.355 | 0.444 | 0.463 | 0.517 | 0.569 |
| 336 | **0.257** | **0.285** | 1.196 | 0.832 | 1.360 | 0.875 | 1.405 | 0.881 | 0.309 | 0.339 | 0.359 | 0.395 | 0.578 | 0.523 | 0.639 | 0.608 |
| 720 | **0.315** | **0.327** | 1.620 | 1.002 | 2.173 | 1.120 | 2.034 | 1.018 | 0.444 | 0.410 | 0.419 | 0.428 | 1.059 | 0.741 | 0.639 | 0.610 |
| Average | **0.330** | **0.347** | 0.533 | 0.482 | 0.654 | 0.575 | 0.731 | 0.591 | 0.487 | 0.468 | 0.394 | 0.415 | 0.819 | 0.661 | 1.008 | 0.663 |

Table 3: Ablation study results on the two parts: **M1** and **M2**.

| Method Metric | Original LaST MSE | LaST MAE | M1 w/o seasonal MSE | w/o seasonal MAE | w/o trend MSE | w/o trend MAE | w/o coe MSE | w/o coe MAE | M2 w/o lower MSE | w/o lower MAE | w/o upper MSE | w/o upper MAE | with MINE MSE | with MINE MAE |
|---|---|---|---|---|---|---|---|---|---|---|---|---|---|---|
| Synthetic 24 | **0.129** | **0.250** | 0.179 | 0.298 | 2.870 | 1.089 | 0.135 | 0.257 | 0.133 | 0.256 | 0.146 | 0.266 | 0.131 | 0.252 |
| 168 | **0.631** | **0.617** | 0.830 | 0.644 | 2.868 | 1.032 | 0.635 | 0.619 | 0.632 | 0.618 | 0.634 | 0.620 | 0.633 | 0.620 |
| 336 | **1.054** | **0.797** | 1.505 | 0.925 | 2.864 | 1.016 | 1.061 | 0.805 | 1.056 | 0.801 | 1.055 | 0.798 | 1.055 | 0.801 |
| ETTh1 24 | **0.324** | **0.368** | 0.328 | 0.373 | 0.631 | 0.548 | 0.334 | 0.380 | 0.339 | 0.383 | 0.332 | 0.376 | 0.332 | 0.375 |
| 48 | **0.351** | **0.380** | 0.360 | 0.389 | 0.973 | 0.695 | 0.358 | 0.386 | 0.360 | 0.389 | 0.361 | 0.390 | 0.358 | 0.387 |
| 168 | **0.468** | **0.453** | 0.482 | 0.463 | 1.106 | 0.788 | 0.497 | 0.474 | 0.508 | 0.482 | 0.477 | 0.459 | 0.502 | 0.484 |
| 336 | **0.566** | **0.512** | 0.579 | 0.531 | 1.206 | 0.843 | 0.598 | 0.546 | 0.604 | 0.543 | 0.582 | 0.534 | 0.603 | 0.545 |
| 720 | **0.740** | **0.650** | 0.788 | 0.672 | 1.240 | 0.854 | 0.803 | 0.689 | 0.766 | 0.665 | 0.768 | 0.670 | 0.780 | 0.669 |
| Average | **0.533** | **0.503** | 0.631 | 0.537 | 1.720 | 0.858 | 0.553 | 0.520 | 0.550 | 0.517 | 0.544 | 0.514 | 0.549 | 0.517 |

Notably, though Autofomer with the simple moving average block achieves satisfying decomposition from the time series perspective, their representations are still prone to entanglement. These results suggest that (1) learning disentangled seasonal-trend representations is not trivial, and (2) the proposed decomposition mechanisms successfully disentangle the seasonal-trend representations in latent space, each paying attention to a specific temporal pattern.

**Input settings.** We further investigate the influence of hyperparameter input length to validate the sensitivity and Table 4 shows the results. Long look-back window improves the performance especially in long-term forecasting, while others even have performance degradation. This verifies that LaST can effectively utilize past information to understand patterns and make predictions.

Table 4: Multivariate forecasting performance with different input lengths on ETTm1 datasets.

| Input length | | 96 | | | 168 | | | 201 | | | 672 | | Average |
|---|---|---|---|---|---|---|---|---|---|---|---|---|---|
| Output length | | 48 | 288 | 672 | 48 | 288 | 672 | 48 | 288 | 672 | 48 | 288 | 672 | |
| LaST | MSE | **0.321** | **0.427** | **0.532** | **0.286** | **0.401** | **0.516** | **0.280** | **0.392** | **0.491** | **0.279** | **0.373** | **0.463** | **0.397** |
| | MAE | **0.359** | **0.421** | **0.493** | **0.338** | **0.413** | **0.493** | **0.329** | **0.403** | **0.466** | **0.334** | **0.391** | **0.448** | **0.407** |
| CoST | MSE | 0.390 | 0.514 | 0.644 | 0.335 | 0.467 | 0.611 | 0.331 | 0.472 | 0.620 | 0.349 | 0.474 | 0.612 | 0.485 |
| | MAE | 0.422 | 0.507 | 0.585 | 0.392 | 0.484 | 0.569 | 0.386 | 0.486 | 0.574 | 0.399 | 0.486 | 0.567 | 0.488 |
| AutoFormer | MSE | 0.454 | 0.634 | 0.606 | 0.460 | 0.578 | 0.529 | 0.463 | 0.545 | 0.562 | 0.562 | 0.656 | 0.695 | 0.562 |
| | MAE | 0.453 | 0.528 | 0.542 | 0.456 | 0.514 | 0.504 | 0.452 | 0.512 | 0.516 | 0.503 | 0.571 | 0.586 | 0.511 |

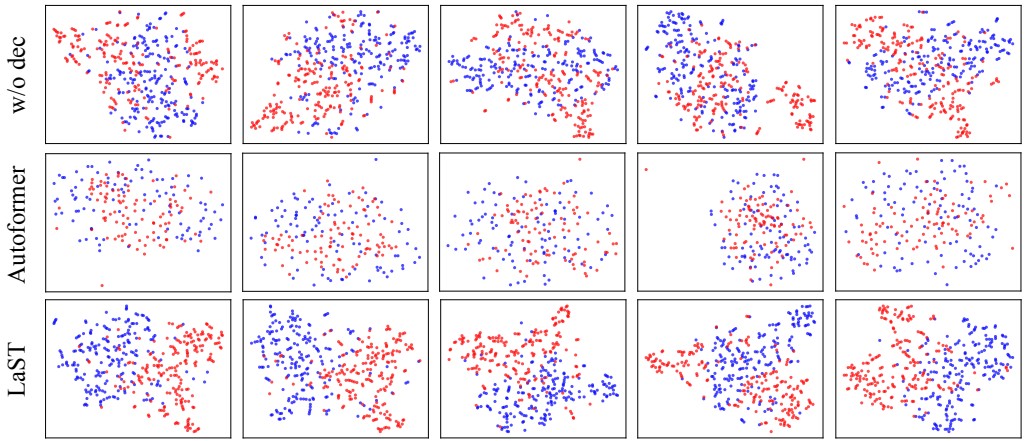

Figure 2: Visualizations of seasonal (red) and trend (blue) representations on ETTh1 dataset.

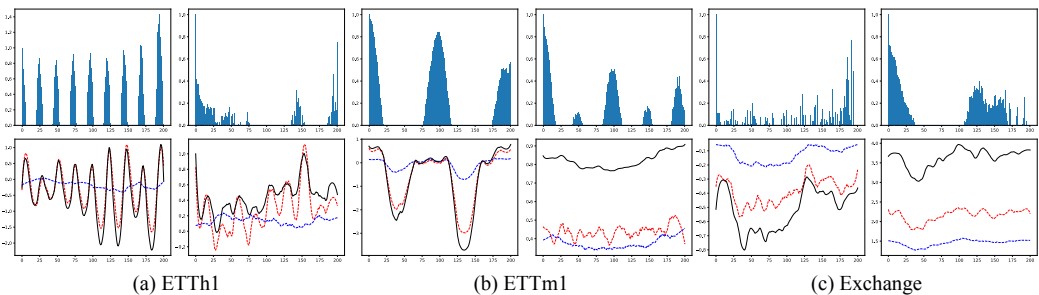

(a) ETTh1        (b) ETTm1        (c) Exchange

Figure 3: *Top*: learned seasonality visualizations (autocorrelation statistics of reconstructed seasonal sequences). *Bottom*: seasonal (red) and trend (blue) reconstructions to the ground truths (black).

**Observations from a case-study.** We further validate LaST by by visualizing the extracted seasonality and trend in specific cases. As shown in Figure 3, LaST can capture the seasonal patterns on real-world datasets. For example, a strong daily period is indicated on hourly and 15-minutes ETT datasets. Even though the period on Exchange dataset is not obvious, LaST still provides some long-term periods on the daily data. Besides, trend and seasonal components jointly accurately restore the original sequence with their own perspective, which supports that LaST can produce workable disentangled representations for intricate time series.

## 6 Conclusion

We presented LaST, a disentangled variational inference framework with mutual information constraints to disassociate a couple of seasonal-trend representations in latent space, for effective forecasting of time series. Our extensive experiments demonstrated that LaST successfully disentangles the seasonal-trend representations and achieves state-of-the-art performance. Our future work will focus on tackling other challenging downstream tasks in the time series domain, e.g., generation and imputation. In addition, we plan to model stochastic factors explicitly in decomposition strategies, which will better understand the real-world time series.

## Acknowledgments and Disclosure of Funding

This work was supported in part by National Natural Science Foundation of China (Grant No.62072077 and No.62176043), Natural Science Foundation of Sichuan Province (Grant No. 2022NSFSC0505), and National Science Foundation SWIFT (Grant No.2030249).

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
