# Learning Latent Seasonal-Trend Representations for Time Series Forecasting (Technical Appendix)

In this body of supplementary materials, we first provide the proofs of ELBO inference with decomposition, reconstruction loss alternative, and MINE biased gradient. Next, we discuss the use of Discrete Fourier transform and autocorrelation methods in our work. This is followed by a more detailed discussion of the use of prior distribution, mutual information upper bounds, along with the complexity results of our algorithmic solutions. Lastly, we present the details of the experimental settings and provide some additional results to support the claims in the main text.

## A  Proofs

Following are the details of the proofs of the claims from the main text – Theorem 1, Theorem 2 and the MI lower bound (cf. Section 4).

### A.1  Inference of ELBO with decomposition

We follow the decomposition strategy, in the sense that time series and their representations are composed of multiple components such as seasonal $Z^s$ and trend $Z^t$. Sequence $X$ and $Y$ is decomposed as a sum form, i.e., $X = X^s + X^t$, $Y = Y^s + Y^t$. Representations $Z^s$ and $Z^t$ are independent from each other, i.e., $P(Z^s, Z^t) = P(Z^s)P(Z^t)$ and, additionally, $Z^t$ and $Z^s$ are only associated with their own components. With the above assumptions, we have:

$$
\begin{aligned}
\mathcal{L}_{ELBO} &= \log \int_Z P_\psi(Y|Z)Q_\phi(Z|X)dZ + \mathbb{E}_{Q_\phi(Z|X)}[\log P_\theta(X|Z)] - \mathbb{KL}(Q_\phi(Z|X)||P(Z)) \\
&= \log \int_{Z^s} \int_{Z^t} P_\psi(Y|Z^s, Z^t)Q_{\phi^s,\phi^t}(Z^s, Z^t|X)dZ^s dZ^t \\
&\quad + \mathbb{E}_{Q_{\phi^s,\phi^t}(Z^s, Z^t|X)}[\log P_\theta(X|Z^s, Z^t)] - \mathbb{KL}(Q_{\phi^s,\phi^t}(Z^s, Z^t|X)||P(Z^s, Z^t)) \\
&= \log \int_{Z^s} \int_{Z^t} P_\psi(Y|Z^s, Z^t)Q_{\phi^s,\phi^t}(Z^s, Z^t|X)dZ^s dZ^t \\
&\quad + \mathbb{E}_{Q_{\phi^s}(Z^s|X)}[\log P_\theta(X|Z^s)] + \mathbb{E}_{Q_{\phi^s}(Z^t|X)}[\log P_\theta(X|Z^t)] \\
&\quad - \mathbb{KL}(Q_{\phi^s}(Z^s|X)||P(Z^s)) - \mathbb{KL}(Q_{\phi^t}(Z^t|X)||P(Z^t)) \\
&= \log \int_Z P_\psi(Y|Z^t, Z^s)Q_{\phi^s,\phi^t}(Z^s, Z^t|X)dZ^s dZ^t \\
&\quad + \mathbb{E}_{Q_{\phi^s}(Z^s|X)}[\log P_{\theta^s}(X^s|Z^s)] + \mathbb{E}_{Q_{\phi^t}(Z^t|X)}[\log P_{\theta^t}(X^t|Z^t)] \\
&\quad - \mathbb{KL}(Q_{\phi^s}(Z^s|X)||P(Z^s)) - \mathbb{KL}(Q_{\phi^t}(Z^t|X)||P(Z^t)). \tag{A1}
\end{aligned}
$$

### A.2  Proof of the alternative of reconstruction loss

The loss function of $\log P_\theta(X|Z)$ can be written with Gaussian distribution as:

$$
-\log P_\theta(X|Z) = \left( \log \sigma_\theta(Z) + \frac{1}{2}\log 2\pi + \frac{1}{2}\frac{\|X - \mu_\theta(Z)\|^2}{\sigma_\theta(Z)} \right) \propto \|X - \mu_\theta(Z)\|^2, \tag{A2}
$$

where $\mu_\theta(Z)$ and $\sigma_\theta(Z)$ are neural networks that reconstruct the $X$ from latent representations. By the above equation, maximizing the reconstruction loss is regarded as minimizing the euclidean distance between inputs and reconstructions.

Thus, we opt to optimize the reconstruction loss in LaST by finding a way that imposes closer distances between raw input $X^s$, $X^t$, and reconstructed input $\hat{X}^s$, $\hat{X}^t$. There are three terms in Eq. (6) in main text:

$$\mathcal{L}_{rec} = -\sum_{\kappa=1}^{T-1} \left\| \mathcal{A}_{XX}(\kappa) - \mathcal{A}_{\hat{X}^s\hat{X}^s}(\kappa) \right\|^2 + \text{CORT}(X, \hat{X}^t) - \left\| \hat{X}^t + \hat{X}^s - X \right\|^2,$$

1. The first term reflects the similarity between every instant value in $X$ and its $\kappa$-lagged value. We can regard the function $\mathcal{A}_{XX}(\kappa)$ as an unnormalized score that measures the confidence of period $\kappa$ [1, 2]. A lower distance between the two autocorrelation sequences means more similar periodic characteristics. With identical autocorrelations, reconstructed $\hat{X}^s$ will be same as $X^s$ in the seasonal view but still holds interference. The optimized reconstructed $\hat{X}^s$ can be formulated as $\hat{X}^s = X^s + \varepsilon^s$, where $\varepsilon$ denotes the interference error.

2. The second normalized term reflects the similarity between the first differences of the input time series, which further measures the simultaneity of two signals' rising or falling patterns, i.e., the trend. With this objective, the reconstructed trend input is formulate as $\hat{X}^t = X^t + \varepsilon^t$ similar to $\hat{X}^s$.

3. The third term now naturally becomes $\|\varepsilon^s + \varepsilon^t\|^2$ aiming to minimize the interference terms.

These three terms are used to reconstruct the pure seasonal and trend components and optimize the above reconstruction loss.

### A.3 Biased gradient of MINE

The lower bound of MI between $X$ and $Z$ is estimated by MINE [3], which is defined as

$$I(X, Z) \geq \mathbb{E}_{Q_\phi(X,Z)}[\gamma_\alpha(X, Z)] - \log \mathbb{E}_{Q(x)Q_\phi(z)}[\exp(\gamma_\alpha(X, Z))] = I_{MINE}. \tag{A3}$$

When sampling the time series in a mini-batch $\mathcal{B}$ (see details in Algorithm 1), it becomes

$$I_{MINE}(X, Z) = \frac{1}{\mathcal{B}} \sum_{i=j=1}^{\mathcal{B}} \gamma_\alpha(X^{(i)}, Z^{(i)}) - \log \frac{1}{\mathcal{B}} \sum_{i=j=1}^{\mathcal{B}} \exp(\gamma_\alpha(X^{(i)}, Z^{(j)})), \tag{A4}$$

where $Z^{(i)}$ and $Z^{(j)}$ are derived from parameters-driven encoders $Q_\phi(Z|X)$ with input $X$. The gradient of MINE is calculated by general stochastic optimizer such as Adam [4] or others:

$$G_\mathcal{B} = \frac{1}{\mathcal{B}} \sum_{i=j=1}^{\mathcal{B}} \nabla_{\alpha,\phi} \gamma_\alpha(X^{(i)}, Z^{(i)}) - \frac{\sum_{i=j=1}^{\mathcal{B}} \nabla_{\alpha,\phi} \gamma_\alpha(X^{(i)}, Z^{(j)}) \exp(\gamma_\alpha(X^{(i)}, Z^{(j)})}{\sum_{i=j=1}^{\mathcal{B}} \exp(\gamma_\alpha(X^{(i)}, Z^{(j)})}, \tag{A5}$$

where the second term holds bias since the derivative of logarithmic function.

## B  Discrete Fourier transform and autocorrelation

The details of our use of Discrete Fourier Transform and efficient autocorrelation are presented next.

### B.1 Discrete Fourier Transform

We apply discrete Fourier Transform (DFT) and inverse Discrete Fourier Transform (iDFT) to construct the mapping between time and frequency domain (cf. Figure 1 in the main text). First, given a time domain sequence of seasonality $Z^s = \{Z_1^s, Z_2^s, \dots, Z_T^s\}$, we consider the first $\lfloor \frac{T+1}{2} \rfloor$ Fourier coefficients according to the Nyquist theorem [5]:

$$Z_{\mathcal{F},k}^s = \mathcal{F}(Z^s)_k = \sum_{t=1}^{T} Z_t^s \cdot \exp(\frac{-2\pi ikt}{T}), \tag{B1}$$

where $1 \leq k \leq \lfloor \frac{T+1}{2} \rfloor$. Then time domain seasonal representation is extended and mapped from frequency domain with iDFT:

$$Z_t^s = \mathcal{F}^{-1}(Z_{\mathcal{F}^s})_t = \frac{1}{T} \sum_{k=1}^{T} Z_{\mathcal{F},k}^s \cdot \exp(\frac{2\pi i k t}{T}), \text{ where } 1 \leq t \leq T + \tau, \tag{B2}$$

where we derive predictable representations $\tilde{Z}^s$ by taking the last $\tau$ time steps. In LaST, we use the fast version of DFT [6] based on halving lemma and divide-and-conquer strategy, based on which the time complexity becomes $\mathcal{O}(T \log T)$, as opposed to the original $\mathcal{O}(T^2)$.

## B.2 Efficient autocorrelation

The autocorrelation of sequence $X$ with $\kappa$ lagged steps is defined as

$$\mathcal{A}_{XX}(\kappa) = \sum_{i=1}^{T-\kappa}(X_t - \bar{X})(X_{t+\kappa} - \bar{X}). \tag{B3}$$

To obtain the autocorrelation sequence, it requires $T(T-1)/2$ times multiplications, each of which holds $\mathcal{O}(d^2)$ time complexity, where $d$ is the dimension of $X$. To improve the efficiency, we employ the Wiener–Khinchin theorem [7] that allows computing the autocorrelation with fast Fourier transform (FFT) and its inverse transform as:

$$X_{\mathcal{F}} = FFT(X^s), \tag{B4}$$
$$S_{\mathcal{F}} = X_{\mathcal{F}} X_{\mathcal{F}}^*, \tag{B5}$$
$$\mathcal{A}_{XX}(0 : T - 1) = iFFT(S_{\mathcal{F}}), \tag{B6}$$

where the asterisk denotes complex conjugate, and the output $\mathcal{A}_{XX}(0 : T - 1)$ denotes sequence $\{\mathcal{A}_{XX}(\kappa)\}_{\kappa \in [0:T-1]}$. In this setting, the number of multiplication calculations drops to $T \log T$ (from the original quadratic complexity).

## C   Discussions of prior distributions

Choosing an appropriate prior for the latent representations in VAE is important for mediating between the encoder and decoder. The standard Gaussian $\mathcal{N}(0, I)$ is a commonly used prior assumption because: (1) it is efficient without additional calculation overhead and (2) the input data are preprocessed with standard normalization. We note that any other advanced approaches on prior learning can be easily incorporated into our model. In the following, we take VampPrior [8] – which produces prior distribution through the encoder $Q_\phi$ given pseudo inputs – as an example to show the effects of different priors on the forecasting performance of LaST. The priors $P(Z^s)$ and $P(Z^t)$ are obtained by $Q_{\phi^s}(Z^s|I)$ and $Q_{\phi^t}(Z^t|I)$, respectively, with identity matrix as pseudo inputs.

Table C1: Multivariate forecasting performance with different prior distributions.

| Dataset | | ETTh1 | | | ETTm1 | | | Exchange | | | Weather | | |
|---|---|---|---|---|---|---|---|---|---|---|---|---|---|
| | | 48 | 336 | 720 | 48 | 288 | 672 | 48 | 336 | 720 | 48 | 336 | 720 |
| Standard Gaussian | MSE | 0.351 | **0.566** | **0.740** | **0.280** | 0.392 | 0.491 | 0.056 | 0.430 | **1.521** | **0.131** | 0.257 | 0.315 |
| | MAE | 0.380 | **0.512** | **0.650** | 0.329 | 0.403 | **0.466** | 0.162 | **0.482** | 0.898 | **0.174** | 0.285 | 0.327 |
| VampPrior | MSE | **0.349** | 0.567 | 0.765 | 0.284 | **0.390** | **0.490** | **0.054** | **0.426** | 1.763 | 0.137 | **0.250** | **0.313** |
| | MAE | **0.377** | 0.522 | 0.666 | 0.337 | 0.411 | 0.468 | **0.161** | 0.484 | 0.927 | 0.196 | **0.279** | **0.326** |

Table C1 shows the experimental results. We can see that VampPrior and standard Gaussian are well matched in LaST on time series forecasting task. However, none shows clear superiority over the other on four datasets. Thus, we use the standard Gaussian as the default prior in LaST due to its efficiency.

## D   Existing upper bounds and the proposed $I_{STUB}$

We now discuss in detail the portion pertaining to the upper bounds (cf. Section 4 of the main paper).

### D.1 Existing upper bounds

Existing research results have concluded several traceable and appropriate upper bounds to MI between $X$ and $Y$ [9, 10, 11], they include:

$$I_{VUB}(X,Y) = \mathbb{E}_{P(X,Y)}\left[\log \frac{P(Y|X)}{Q(Y)}\right], \tag{D1}$$

$$I_{L1Out}(X,Y) = \mathbb{E}\left[\frac{1}{\mathcal{B}}\sum_{i=1}^{\mathcal{B}}\left[\log \frac{P(Y^{(i)}|X^{(i)})}{\frac{1}{\mathcal{B}-1}\sum_{j\neq i}P(Y^{(i)}|X^{(j)})}\right]\right], \tag{D2}$$

$$I_{CLUB}(X,Y) = \mathbb{E}_{P(X,Y)}[\log P(Y|X)] - \mathbb{E}_{P(X)P(Y)}[\log P(Y|X)], \tag{D3}$$

where $Q(Y)$ is usually a learned marginal density approximation to $P(Y)$ and $\mathcal{B}$ is the batch size. These upper bounds are all based on conditional distributions which, defined as $Q(Z^s|Z^t)$ or $Q(Z^t|Z^s)$ in our LaST, are still untraceable and ineffective. Though we can replace them with joint distribution via deductions, for example,

$$I(Z^s, Z^t) \leq \mathbb{E}_{Q(Z^s, Z^t)}[\log Q(Z^s|Z^t) + \log Q(Z^t)] - \mathbb{E}_{Q(Z^s)Q(Z^t)}[\log Q(Z^s|Z^t) + \log Q(Z^t)]$$
$$= \mathbb{E}_{Q(Z^s, Z^t)}[\log Q(Z^s, Z^t)] - \mathbb{E}_{Q(Z^s)Q(Z^t)}[\log Q(Z^s, Z^t)], \tag{D4}$$

it is challenging to establish a joint distribution and estimate its density directly without any additional information [12].

### D.2 Derivation of proposed upper bound $I_{STUB}$

To overcome the limitations of the existing upper bounds, we introduce a normalized energy-based function that uses a critic $\gamma_\beta(Z^s, Z^t)$ over the variational joint distribution to estimate the probability:

$$Q(Z^s, Z^t) = \frac{Q(Z^s)Q(Z^t)}{\mathcal{Z}_\beta}e^{\gamma_\beta(Z^s, Z^t)}, \text{ where } \mathcal{Z}_\beta = \mathbb{E}_{Q(Z^s)Q(Z^t)}[e^{\gamma_\beta(Z^s, Z^t)}]. \tag{D5}$$

Applying this formula into Eq. (D4), we can obtain a traceable upper bound as:

$$I(Z^s, Z^t) \leq \mathbb{E}_{Q(Z^s, Z^t)}[\log Q(Z^s) + \log Q(Z^t) + \gamma_\beta(Z^s, Z^t) - \log \mathcal{Z}_\beta]$$
$$- \mathbb{E}_{Q(Z^s)Q(Z^t)}[\log Q(Z^s) + \log Q(Z^t) + \gamma_\beta(Z^s, Z^t) - \log \mathcal{Z}_\beta]$$
$$= \mathbb{E}_{Q(Z^s, Z^t)}[\gamma_\beta(Z^s, Z^t)] - \mathbb{E}_{Q(Z^s)Q(Z^t)}[\gamma_\beta(Z^s, Z^t)] = I_{STUB}, \tag{D6}$$

which helps us addressing the joint density calculation challenge.

# E    Complexity analysis and algorithm

This section presents the complexity analysis and, for completeness, also shows the pseudo-code of the algorithms for training and prediction phases.

## E.1    Complexity analysis

The corresponding time complexities of the components in LaST and are shown in Table E1. The time complexity of seasonal and trend encoders/decoders are linear of fully connected network, i.e., $\mathcal{O}(Tnd)$, where $n$ is the number of observations and $d$ is the dimension of latent representation. For the reconstruction phrase, calculations of autocorrelation and CORT incur $\mathcal{O}(nT \log T)$ and $\mathcal{O}(nT)$ time complexities, respectively. The MI upper and lower bounds are both based on the energy function implemented by a 2-layer MLP, which has a time complexity of $\mathcal{O}(Td^2)$. As for the predictor, the time complexity of fast Fourier transform is equal to $\mathcal{O}(nT \log T)$. The first FFN in trend forecasting consumes $\mathcal{O}(nT\tau)$, while the time complexity of other FFNs becomes $\mathcal{O}(\tau nd)$. We note that only the encoders and predictor are required after training. Thus, the time complexity of training is $\mathcal{O}(T(nd + n \log T + d^2) + \tau(nT + nd))$ and that of prediction turns to $\mathcal{O}(nT\tau + nT \log T + \tau nd)$.

Table E1: Complexity analysis of main components in LaST.

| Component | Training | Prediction | Time Complexity |
|---|---|---|---|
| seasonal-trend encoder | ✓ | ✓ | $\mathcal{O}(Tnd)$ |
| seasonal-trend decoder | ✓ | ✗ | $\mathcal{O}(Tnd)$ |
| reconstruction | ✓ | ✗ | $\mathcal{O}(nT \log T)$ |
| MI estimation | ✓ | ✗ | $\mathcal{O}(Td^2)$ |
| seasonal predictor | ✓ | ✓ | $\mathcal{O}(nT \log T + \tau nd)$ |
| trend predictor | ✓ | ✓ | $\mathcal{O}(nT\tau + \tau nd)$ |

## E.2    Algorithm of LaST

The pseudo-codes of LaST covering training and prediction phases are summarized in Algorithm E1 and E2, respectively.

---

**Algorithm E1** Training phase of LaST.

---

**Input:**
    Historical time series $X_{1:T}$ and future time series $X_{T+1:T+\tau}$;
    Hyperparameters and initialized parameters of LaST.
**Output:** LaST with optimized parameters.
  1: **for** a mini-batch with size $\mathcal{B}$ consisting of $\{X^{(i)}, Y^{(i)}\}_{i \in \mathcal{B}}$ **in** training set **do**
  2:     Encode historical time series into latent seasonal-trend representations $Z^s$ and $Z^t$;
  3:     Reconstruct seasonal and trend $\hat{X}^s$ and $\hat{X}^t$ from representations via decoders;
  4:     Calculate the reconstruction loss $\mathcal{L}_{rec}$ via autocorrelation and CORT (cf. Eq. (6));
  5:     Estimate the mutual information bounds via Eq. (9) and Eq. (11);
  6:     Derive the forecasting $Y = \hat{X}_{T+1:T+\tau}$ from $Z^s$ and $Z^t$ via predictor;
  7:     Calculate the MAE loss between $\hat{X}_{T+1:T+\tau}$ and $X_{T+1:T+\tau}$;
  8:     Update parameters of LaST via Adam optimizer;
  9: **end** *until convergence*

---

---

**Algorithm E2** Prediction phase of LaST.

---

**Input:** Historical time series $X_{1:T}$ and optimized LaST;
**Output:** Predicted future time series $\hat{X}_{T+1:T+\tau}$.
  1: Encode historical time series into latent seasonal-trend representations $Z^s$ and $Z^t$;
  2: Derive the forecasting $\hat{X}_{T+1:T+\tau}$ from $Z^s$ and $Z^t$ via predictor;

---

# F Experiment supplementary

In this section, we present more details of the baselines, computing infrastructure, dataset generation process, and full experimental results on ETT benchmark (ETTh1, h2, m1, and m2). In addition, we provide fluctuation analysis and memory cost comparison.

## F.1 Details of baselines

The implementations and settings of the seven baselines are provided below. Unless otherwise specified, we use the suggested settings described in the respective papers.

**CoST [13]:** analyses raw signals in time and frequency domain with temporal convolutions and Fourier transform, and separately produces seasonal and trend representations for time series forecasting task. We run experiments with their publicly available code: `https://github.com/salesforce/CoST`.

**TS2Vec [14]:** is a universal time series representation learning framework that performs contrastive learning in a hierarchical way over augmented context views, which enables a robust contextual representation for each timestamp. We run the code from their open source repository: `https://github.com/yuezhihan/ts2vec`.

**TNC [15]:** is a self-supervised framework that introduces the concept of a temporal neighborhood with stationary properties and learns generalizable time series representations. We use their open-source code: `https://github.com/sanatonek/TNC_representation_learning`. As for hyperparameters, we set $w = 0.005$ in the loss function, $mc\_sample\_size = 20$, batch size as 8, and learning rate as $10^{-3}$ with Adam optimizer.

**VAE-GRU [16]:** combines the strengths of recurrent network and stochastic gradient variational Bayes, which has been applied for time series forecasting among various domains [17, 18]. We implement it by the GRU [19] variant with initial state sampling from $\mathcal{N}(0, I)$ and keep the variational settings consistent with our LaST.

**Autoformer [20]:** designs a novel decomposition architecture with an autocorrelation mechanism to improve Transformer [21]. It exploits average pooling strategy to split time series into seasonal and trend components and regards autocorrelation coefficient as attention score, which boosts the performance of long-term forecasting. We use their publicly available code: `https://github.com/thuml/Autoformer`.

**Informer [22]:** designs an efficient transformer-based model for time series forecasting with a ProbSparse self-attention mechanism, highlights distilling module, and generative style decoder. We use the publicly available code: `https://github.com/zhouhaoyi/Informer2020`.

**TCN [23]:** introduces an architecture that is applied across all tasks for sequence modeling. It is also popular in time series modeling and forecasting [24, 25]. We use their public repository: `https://github.com/locuslab/TCN` and set kernel sizes as $\{2^1, 2^2, \cdots, 2^7\}$.

## F.2 Evaluation environment

LaST is implemented by `PyTorch` and all experiments are run on a server with a `Intel Xeon Platinum 8124M` CPU, a `RTX-3090` GPU, and 128GB memory.

## F.3 Synthetic dataset generation

Following the method in [13], we synthesize time series with sinusoidal seasonal patterns, as well as linear and non-linear trend patterns. The seasonal signal consists of three sine waves with the following period, phase, and amplitudes: $\{(20, 3, 0), (50, 3, 0.2), (100, 3, 0.5)\}$. The trend signal is composed of two components: (1) a nonlinear, saturating pattern $X_t = \frac{1}{1+\exp \beta_0(t-\beta_1)}$, where $\beta_0 = 0.2$ and $\beta_1 = 200$; (2) an ARMA process whose parameters of AR and MA are $\{(0.9, -0.1), (0.2, -0.5)\}$. The final time series is generated by the sum of these seasonal and trend signals.

### F.4 More results on ETT benchmark

Table F1: Performance comparisons on univariate forecasting on complete ETT benchmark. Best performance is highlighted in bold font and the second best results are underlined.

| Method | | LaST | | CoST | | TS2Vec | | TNC | | VAE-GRU | | Autoformer | | Informer | | TCN | |
|---|---|---|---|---|---|---|---|---|---|---|---|---|---|---|---|---|---|
| | | MSE | MAE | MSE | MAE | MSE | MAE | MSE | MAE | MSE | MAE | MSE | MAE | MSE | MAE | MSE | MAE |
| ETTh1 | 24 | **0.030** | **0.131** | 0.040 | 0.152 | 0.039 | 0.151 | 0.057 | 0.184 | 0.042 | 0.155 | _0.057_ | _0.189_ | 0.098 | 0.247 | 0.104 | 0.254 |
| | 48 | **0.051** | **0.169** | _0.060_ | _0.186_ | 0.062 | 0.189 | 0.094 | 0.239 | 0.077 | 0.218 | 0.070 | 0.207 | 0.158 | 0.319 | 0.206 | 0.366 |
| | 168 | **0.078** | **0.211** | _0.097_ | _0.236_ | 0.142 | 0.291 | 0.171 | 0.329 | 0.172 | 0.344 | 0.108 | 0.260 | 0.183 | 0.346 | 0.462 | 0.586 |
| | 336 | **0.100** | **0.246** | _0.112_ | _0.258_ | 0.160 | 0.316 | 0.179 | 0.345 | 0.140 | 0.301 | 0.119 | 0.281 | 0.222 | 0.387 | 0.422 | 0.564 |
| | 720 | _0.138_ | _0.298_ | 0.148 | 0.306 | 0.179 | 0.345 | 0.235 | 0.408 | 0.204 | 0.381 | **0.109** | **0.264** | 0.269 | 0.435 | 0.438 | 0.578 |
| ETTh2 | 24 | **0.070** | **0.197** | 0.079 | 0.207 | 0.097 | 0.230 | 0.097 | 0.238 | _0.073_ | _0.202_ | 0.112 | 0.259 | 0.093 | 0.240 | 0.109 | 0.251 |
| | 48 | **0.099** | **0.239** | 0.118 | 0.259 | 0.124 | 0.274 | 0.131 | 0.281 | _0.110_ | _0.252_ | 0.122 | 0.269 | 0.155 | 0.314 | 0.147 | 0.302 |
| | 168 | **0.169** | **0.321** | 0.189 | 0.339 | 0.198 | 0.355 | 0.197 | 0.354 | 0.179 | 0.340 | _0.178_ | _0.331_ | 0.232 | 0.389 | 0.209 | 0.366 |
| | 336 | **0.202** | _0.362_ | _0.206_ | **0.360** | 0.205 | 0.364 | 0.207 | 0.366 | 0.219 | 0.370 | 0.236 | 0.386 | 0.263 | 0.417 | 0.237 | 0.391 |
| | 720 | 0.247 | 0.404 | 0.214 | 0.371 | 0.208 | 0.371 | _0.207_ | _0.370_ | 0.294 | 0.439 | 0.284 | 0.427 | 0.277 | 0.431 | **0.200** | **0.367** |
| ETTm1 | 24 | **0.011** | **0.077** | _0.015_ | _0.088_ | 0.016 | 0.093 | 0.019 | 0.103 | 0.013 | 0.082 | 0.022 | 0.115 | 0.030 | 0.137 | 0.027 | 0.127 |
| | 48 | **0.021** | **0.108** | _0.025_ | _0.117_ | 0.028 | 0.126 | 0.045 | 0.162 | 0.026 | 0.120 | 0.032 | 0.138 | 0.069 | 0.203 | 0.040 | 0.154 |
| | 96 | **0.033** | **0.134** | _0.038_ | _0.147_ | 0.045 | 0.162 | 0.054 | 0.178 | 0.046 | 0.164 | 0.045 | 0.168 | 0.194 | 0.372 | 0.097 | 0.246 |
| | 288 | **0.069** | **0.197** | 0.077 | 0.209 | 0.095 | 0.235 | 0.142 | 0.290 | 0.127 | 0.294 | _0.071_ | _0.207_ | 0.401 | 0.554 | 0.305 | 0.455 |
| | 672 | **0.100** | **0.239** | 0.113 | 0.257 | 0.142 | 0.290 | 0.136 | 0.290 | 0.217 | 0.399 | _0.102_ | _0.254_ | 0.512 | 0.644 | 0.445 | 0.576 |
| ETTm2 | 24 | **0.026** | **0.108** | _0.027_ | _0.112_ | 0.038 | 0.139 | 0.045 | 0.151 | 0.030 | 0.113 | 0.076 | 0.208 | 0.036 | 0.141 | 0.048 | 0.153 |
| | 48 | **0.051** | **0.157** | 0.054 | _0.159_ | 0.069 | 0.194 | 0.080 | 0.201 | _0.052_ | 0.163 | 0.115 | 0.260 | 0.069 | 0.200 | 0.063 | 0.191 |
| | 96 | **0.069** | **0.191** | _0.072_ | _0.196_ | 0.089 | 0.225 | 0.094 | 0.229 | 0.073 | 0.200 | 0.091 | 0.230 | 0.095 | 0.240 | 0.129 | 0.265 |
| | 288 | **0.119** | **0.260** | _0.137_ | _0.279_ | 0.153 | 0.307 | 0.161 | 0.306 | 0.155 | 0.309 | 0.169 | 0.320 | 0.211 | 0.367 | 0.208 | 0.352 |
| | 672 | **0.171** | **0.318** | _0.183_ | _0.329_ | 0.201 | 0.357 | 0.197 | 0.352 | 0.201 | 0.357 | 0.197 | 0.346 | 0.267 | 0.417 | 0.222 | 0.377 |

Table F2: Performance comparisons on multivariate forecasting on complete ETT benchmark. Best performance is highlighted in bold font and the second best results are underlined.

| Method | | LaST | | CoST | | TS2Vec | | TNC | | VAE-GRU | | AutoFormer | | Informer | | TCN | |
|---|---|---|---|---|---|---|---|---|---|---|---|---|---|---|---|---|---|
| | | MSE | MAE | MSE | MAE | MSE | MAE | MSE | MAE | MSE | MAE | MSE | MAE | MSE | MAE | MSE | MAE |
| ETTh1 | 24 | **0.324** | **0.368** | 0.386 | 0.429 | 0.590 | 0.531 | 0.708 | 0.592 | 0.529 | 0.534 | _0.384_ | _0.428_ | 0.577 | 0.549 | 0.583 | 0.547 |
| | 48 | **0.351** | **0.380** | 0.437 | 0.464 | 0.624 | 0.555 | 0.749 | 0.619 | 0.612 | 0.593 | _0.392_ | _0.419_ | 0.685 | 0.625 | 0.670 | 0.606 |
| | 168 | **0.468** | **0.453** | 0.643 | 0.582 | 0.762 | 0.639 | 0.884 | 0.699 | 0.758 | 0.674 | _0.490_ | _0.481_ | 0.931 | 0.752 | 0.811 | 0.680 |
| | 336 | _0.566_ | _0.512_ | 0.812 | 0.679 | 0.931 | 0.728 | 1.020 | 0.768 | 0.844 | 0.692 | **0.505** | **0.484** | 1.128 | 0.873 | 1.132 | 0.815 |
| | 720 | _0.758_ | _0.659_ | 0.970 | 0.771 | 1.063 | 0.799 | 1.157 | 0.830 | 1.045 | 0.816 | **0.498** | **0.500** | 1.215 | 0.896 | 1.165 | 0.813 |
| ETTh2 | 24 | **0.175** | **0.272** | 0.447 | 0.502 | 0.423 | 0.489 | 0.612 | 0.595 | 0.267 | 0.364 | _0.261_ | _0.341_ | 0.720 | 0.665 | 0.935 | 0.754 |
| | 48 | **0.229** | **0.312** | 0.699 | 0.637 | 0.619 | 0.605 | 0.840 | 0.716 | 0.523 | 0.529 | _0.312_ | _0.373_ | 1.457 | 1.001 | 1.300 | 0.911 |
| | 168 | _0.722_ | _0.609_ | 1.549 | 0.982 | 1.845 | 1.074 | 2.359 | 1.213 | 2.519 | 1.301 | **0.457** | **0.455** | 3.489 | 1.515 | 4.017 | 1.579 |
| | 336 | _1.261_ | _0.828_ | 1.749 | 1.042 | 2.194 | 1.197 | 2.782 | 1.349 | 3.589 | 1.577 | **0.471** | **0.475** | 2.723 | 1.340 | 3.460 | 1.456 |
| | 720 | _1.780_ | _1.094_ | 1.971 | 1.092 | 2.636 | 1.370 | 2.753 | 1.394 | 3.788 | 1.683 | **0.474** | **0.484** | 3.467 | 1.473 | 3.106 | 1.381 |
| ETTm1 | 24 | **0.218** | **0.289** | _0.246_ | _0.329_ | 0.453 | 0.444 | 0.522 | 0.472 | 0.509 | 0.534 | 0.383 | 0.403 | 0.453 | 0.444 | 0.522 | 0.472 |
| | 48 | **0.280** | **0.329** | _0.331_ | _0.386_ | 0.592 | 0.521 | 0.695 | 0.567 | 0.642 | 0.543 | 0.454 | 0.453 | 0.494 | 0.503 | 0.542 | 0.508 |
| | 96 | **0.323** | **0.360** | _0.378_ | _0.419_ | 0.635 | 0.554 | 0.731 | 0.595 | 0.600 | 0.540 | 0.481 | 0.463 | 0.678 | 0.614 | 0.666 | 0.578 |
| | 288 | **0.392** | **0.403** | _0.472_ | _0.486_ | 0.693 | 0.597 | 0.818 | 0.649 | 0.769 | 0.678 | 0.634 | 0.528 | 1.056 | 0.786 | 0.991 | 0.735 |
| | 672 | **0.491** | **0.466** | 0.620 | 0.574 | 0.782 | 0.653 | 0.932 | 0.712 | 0.799 | 0.673 | _0.606_ | _0.542_ | 1.192 | 0.926 | 1.032 | 0.756 |
| ETTm2 | 24 | **0.102** | **0.206** | _0.141_ | _0.282_ | 0.179 | 0.296 | 0.185 | 0.297 | 0.178 | 0.306 | 0.153 | 0.261 | 0.173 | 0.301 | 0.180 | 0.324 |
| | 48 | **0.135** | **0.237** | 0.209 | 0.347 | 0.243 | 0.352 | 0.264 | 0.360 | 0.214 | 0.332 | _0.178_ | _0.280_ | 0.303 | 0.409 | 0.204 | 0.327 |
| | 96 | **0.182** | **0.274** | 0.325 | 0.436 | 0.336 | 0.415 | 0.389 | 0.458 | 0.279 | 0.375 | _0.255_ | _0.339_ | 0.365 | 0.453 | 3.041 | 1.330 |
| | 288 | **0.299** | **0.360** | 0.816 | 0.698 | 0.707 | 0.632 | 0.920 | 0.788 | 0.809 | 0.691 | _0.342_ | _0.378_ | 1.047 | 0.804 | 3.162 | 1.337 |
| | 672 | _0.790_ | _0.611_ | 1.633 | 1.025 | 1.801 | 1.022 | 2.164 | 1.135 | 1.838 | 1.038 | **0.434** | **0.430** | 3.126 | 1.302 | 3.624 | 1.484 |

## F.5 Fluctuation analysis

To validate the models' robustness, we conduct fluctuation analysis and have three runs of all the experiments with different seeds for baselines and LaST. Table F3 reports the performance results in terms of MSE and MAE with standard deviations.

Table F3: Standard deviations results on multivariate forecasting.

| Method | | LaST | | CoST | | VAE-GRU | | AutoFormer | |
| | | MSE | MAE | MSE | MAE | MSE | MAE | MSE | MAE |
|---|---|---|---|---|---|---|---|---|---|
| ETTh1 | 24 | **0.324** ±0.003 | **0.368** ±0.004 | 0.386 ±0.006 | 0.429 ±0.005 | 0.529 ±0.016 | 0.531 ±0.013 | 0.384 ±0.020 | 0.428 ±0.017 |
| | 48 | **0.351** ±0.002 | **0.380** ±0.003 | 0.437 ±0.003 | 0.464 ±0.006 | 0.612 ±0.019 | 0.593 ±0.010 | 0.392 ±0.020 | 0.419 ±0.026 |
| | 168 | **0.468** ±0.004 | **0.453** ±0.004 | 0.643 ±0.018 | 0.582 ±0.011 | 0.758 ±0.035 | 0.647 ±0.031 | 0.490 ±0.016 | 0.481 ±0.009 |
| | 336 | 0.566 ±0.002 | 0.512 ±0.003 | 0.812 ±0.003 | 0.679 ±0.008 | 0.844 ±0.030 | 0.692 ±0.028 | **0.505** ±0.013 | **0.484** ±0.015 |
| | 720 | 0.758 ±0.001 | 0.659 ±0.001 | 0.970 ±0.030 | 0.771 ±0.010 | 1.045 ±0.014 | 0.816 ±0.015 | **0.498** ±0.011 | **0.500** ±0.012 |
| ETTm1 | 24 | **0.218** ±0.002 | **0.289** ±0.001 | 0.246 ±0.018 | 0.329 ±0.010 | 0.509 ±0.023 | 0.402 ±0.027 | 0.383 ±0.030 | 0.403 ±0.035 |
| | 48 | **0.280** ±0.003 | **0.329** ±0.005 | 0.331 ±0.024 | 0.386 ±0.008 | 0.642 ±0.026 | 0.543 ±0.028 | 0.454 ±0.029 | 0.453 ±0.038 |
| | 96 | **0.323** ±0.001 | **0.360** ±0.001 | 0.378 ±0.023 | 0.419 ±0.012 | 0.600 ±0.022 | 0.540 ±0.020 | 0.481 ±0.018 | 0.463 ±0.020 |
| | 288 | **0.392** ±0.003 | **0.403** ±0.027 | 0.472 ±0.011 | 0.486 ±0.007 | 0.769 ±0.038 | 0.678 ±0.019 | 0.634 ±0.024 | 0.528 ±0.016 |
| | 672 | **0.491** ±0.006 | **0.466** ±0.002 | 0.620 ±0.029 | 0.574 ±0.010 | 0.799 ±0.008 | 0.673 ±0.010 | 0.606 ±0.010 | 0.542 ±0.014 |
| Electricity | 24 | **0.125** ±0.002 | **0.222** ±0.003 | 0.136 ±0.006 | 0.242 ±0.005 | 0.190 ±0.004 | 0.250 ±0.002 | 0.165 ±0.002 | 0.286 ±0.004 |
| | 48 | **0.146** ±0.001 | **0.245** ±0.001 | 0.153 ±0.002 | 0.258 ±0.003 | 0.228 ±0.003 | 0.280 ±0.005 | 0.178 ±0.007 | 0.295 ±0.010 |
| | 168 | **0.170** ±0.002 | **0.265** ±0.003 | 0.175 ±0.005 | 0.275 ±0.008 | 0.240 ±0.005 | 0.297 ±0.009 | 0.215 ±0.003 | 0.327 ±0.004 |
| | 336 | **0.188** ±0.004 | **0.280** ±0.002 | 0.196 ±0.005 | 0.296 ±0.005 | 0.262 ±0.007 | 0.318 ±0.007 | 0.218 ±0.006 | 0.329 ±0.004 |
| | 720 | **0.223** ±0.003 | **0.309** ±0.002 | 0.232 ±0.008 | 0.327 ±0.003 | 0.296 ±0.004 | 0.347 ±0.003 | 0.252 ±0.007 | 0.356 ±0.008 |
| Exchange | 24 | **0.033** ±0.001 | **0.122** ±0.001 | 0.033 ±0.003 | 0.127 ±0.009 | 0.064 ±0.004 | 0.178 ±0.005 | 0.060 ±0.008 | 0.178 ±0.007 |
| | 48 | **0.056** ±0.001 | **0.162** ±0.002 | 0.058 ±0.002 | 0.165 ±0.004 | 0.133 ±0.007 | 0.262 ±0.010 | 0.091 ±0.010 | 0.222 ±0.009 |
| | 168 | **0.190** ±0.009 | **0.320** ±0.003 | 0.198 ±0.015 | 0.327 ±0.008 | 0.334 ±0.013 | 0.432 ±0.012 | 0.405 ±0.017 | 0.473 ±0.013 |
| | 336 | **0.430** ±0.018 | **0.482** ±0.016 | 0.512 ±0.026 | 0.523 ±0.013 | 0.614 ±0.030 | 0.606 ±0.024 | 0.509 ±0.041 | 0.524 ±0.016 |
| | 720 | 1.521 ±0.162 | **0.898** ±0.105 | 1.855 ±0.103 | 0.998 ±0.037 | 2.285 ±0.078 | 1.117 ±0.054 | **1.447** ±0.084 | 0.941 ±0.028 |
| Weather | 24 | **0.105** ±0.005 | **0.134** ±0.004 | 0.293 ±0.013 | 0.369 ±0.009 | 0.117 ±0.004 | 0.147 ±0.002 | 0.180 ±0.015 | 0.263 ±0.018 |
| | 48 | **0.131** ±0.006 | **0.174** ±0.006 | 0.558 ±0.020 | 0.515 ±0.014 | 0.227 ±0.008 | 0.270 ±0.003 | 0.241 ±0.010 | 0.310 ±0.012 |
| | 168 | **0.197** ±0.010 | **0.238** ±0.011 | 0.812 ±0.033 | 0.671 ±0.026 | 0.234 ±0.007 | 0.280 ±0.005 | 0.295 ±0.027 | 0.355 ±0.023 |
| | 336 | **0.257** ±0.013 | **0.285** ±0.014 | 1.196 ±0.024 | 0.832 ±0.022 | 0.309 ±0.022 | 0.339 ±0.015 | 0.359 ±0.035 | 0.395 ±0.031 |
| | 720 | **0.315** ±0.003 | **0.327** ±0.009 | 1.620 ±0.036 | 1.002 ±0.020 | 0.444 ±0.037 | 0.410 ±0.024 | 0.419 ±0.017 | 0.428 ±0.014 |

## F.6 Convergence

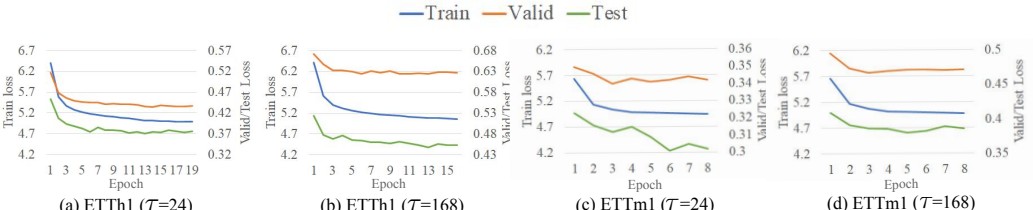

(a) ETTh1 ($\mathcal{T}$=24)   (b) ETTh1 ($\mathcal{T}$=168)   (c) ETTm1 ($\mathcal{T}$=24)   (d) ETTm1 ($\mathcal{T}$=168)

Figure F1: Train, valid, and test losses changes in the training process.

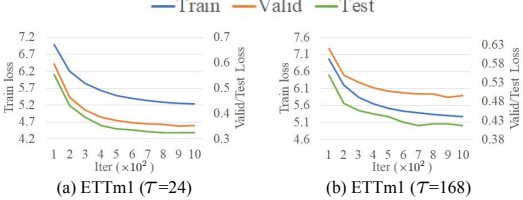

(a) ETTm1 ($\mathcal{T}$=24)   (b) ETTm1 ($\mathcal{T}$=168)

Figure F2: Train, valid, and test losses changes at the first epoch.

We conduct additional experiments to validate the convergence property of our model. Figure F1 shows the descending process of training, valid, and test loss of our model as the epochs increase, where we can see that all losses first drop and then goes to level off. The downward trend is not obvious on ETTm1 dataset. This is because ETTm1 is a larger-scale dataset and it requires over 1,000 times stochastic gradient descent at every epoch. We visualize the losses change at the first

epoch in figure F2. We can observe that losses drop a lot at the first epoch. These results support that our model converges well on real-world datasets.

### F.7 Memory cost

We keep the batch size $\mathcal{B}$, input length $T$ and forecasting length $\tau$ consistent among baselines and LaST, and report the memory cost of in Table F4, where we can observe that LaST is $\sim$30% more memory-efficient than CoST and over 40% more memory-efficient than AutoFormer. CoST contains multiple TCNs with different kernel sizes, which enable it to collect multi-horizon information, but lead the memory consumption to multiply. Transformer-based Autoformer establishes the correlations between every time step, which captures the long-term dependencies but incurs $\mathcal{O}((T+\tau)^2)$ space complexity. For example, AutoFormer on Weather dataset with $\{\mathcal{B}=32, T=201, \tau=720\}$ allocates 9045MB memory while LaST only needs 1751MB of memory. Based on variational inference theory, our proposed LaST model regards the time series as a whole and extracts seasonal and trend without additional memory consumption.

Table F4: Memory usage (MB) comparisons among different methods.

| Batch size | | 16 | | | | | | 32 | | | | | |
|---|---|---|---|---|---|---|---|---|---|---|---|---|---|
| Input length | | | 96 | | | 201 | | | 96 | | | 201 | |
| Output length | | 48 | 168 | 720 | 48 | 168 | 720 | 48 | 168 | 720 | 48 | 168 | 720 |
| LaST | ETTh1 | 1071 | 1073 | 1149 | 1127 | 1139 | 1383 | 1133 | 1175 | 1317 | 1245 | 1403 | 1741 |
| | Weather | 1075 | 1081 | 1175 | 1131 | 1167 | 1395 | 1139 | 1187 | 1337 | 1255 | 1407 | 1751 |
| CoST | ETTh1 | 1597 | 1597 | 1597 | 1909 | 1909 | 1909 | 1759 | 1759 | 1759 | 1977 | 1977 | 1977 |
| | Weather | 1598 | 1598 | 1598 | 1911 | 1911 | 1911 | 1791 | 1761 | 1761 | 1981 | 1981 | 1981 |
| AutoFormer | ETTh1 | 1831 | 2501 | 4371 | 2209 | 2857 | 4771 | 2433 | 3477 | 8213 | 3263 | 4213 | 9035 |
| | Weather | 1833 | 2505 | 4381 | 2211 | 2859 | 4777 | 2437 | 3481 | 8229 | 3265 | 4219 | 9045 |