# OpenReview forum: "Learning Latent Seasonal-Trend Representations for Time Series Forecasting"
_NeurIPS.cc/2022/Conference — NeurIPS 2022 Accept_

### Official Review · Reviewer_GaRG · 2022-07-11

**Rating:** 7
**Confidence:** 4
**Soundness:** 3 good
**Presentation:** 4 excellent
**Contribution:** 3 good

**Summary:**

The author proposed VAE-based method, LaST, to disentangle the seasonal-trend representations of sequential data. The LaST uses the trend and seasonal data as the reconstructed targets, which enforce the model to learn representations dedicated to its target hence achieving better disentanglement. Finally, a predictor is introduced to guarantee its performance on the downstream tasks. Results show that LaST can achieve better predicting performance compared to other forecasting baselines.

**Questions:**

1. Does these trend data Xt and seasonal data Xs are extracted from X? If so, how do you extract them?
2. Can you compare your results with other disentangled representation learning methods for sequential data?


**Limitations:**

1. The generalisability of this LaST might be limited since it requires that the input data are a triplet: Time series data X, trend data Xt and seasonal data Xs. Many sequential datasets, especially the real-world ones, often don't have the complete triplet data or with a lot of missing/noising data. So its generalisability should be further investigated.

2. The author might need to add more baselines (other disentangled representation learning methods for sequential data).

**Strengths And Weaknesses:**

Pros:
1. The usage of trend and seasonal input Xt and Xs force the model to learn dedicated representations and make us easier to evaluate the disentanglement.
2. The author provides rigorous mathematical proof, including the decomposition of ELBO, and the lower and upper bounds for MI optimization.
3. The author provides extensive results on both the prediction performance for downstream tasks and the disentanglement of the trend and seasonal features.

Cons:
1. The author stated that "existing approaches with a single high-dimensional representation sacrifice the information utilization and explainability". Models like CoST uses different modules to extract trend/seasonal dependencies. The author fails to address the difference between those models that do not use a high-dimensional representation.

2. Other disentangled representation learning methods for sequential data are not referenced as baselines i

---

> ### Author Response · Authors · 2022-08-02
> **Response to Reviewer GaRG**
>
> Many Thanks to Reviewer GaRG for the thorough review and valuable comments. The responses to your concerns are as below.
>
> -------
>
> **Q1:** Does these trend data Xt and seasonal data Xs are extracted from X? If so, how do you extract them?
>
> **A1:** Thanks for this constructive comment. Trend data $X^t$ and seasonal data $X^s$ cannot be extracted directly from the $X$. Thus, we use reconstruction $\hat{X}^t$ and $\hat{X}^s$ as substitutes in Eq. (6). Seasonal $\hat{X}^s$ and trend $\hat{X}^t$ are reconstructed by the encoding and decoding process of variational inference. Specifically, as algorithm E1 in Appendix E.2 shows, an input time series $X$ is fed into the seasonal encoder and trend encoder and is encoded as two disentangled representations, i.e., $Z^s$ and $Z^t$. Then, the decoders reconstruct them into $\hat{X}^s$ and $\hat{X}^t$ for further estimations and computations. Hope our explanation helps. We will explain the construction process more clearly in the main text.
>
> -------
>
> **Q2:** The generalizability of this LaST might be limited since it requires that the input data are a triplet: Time series data X, trend data Xt and seasonal data Xs. Many sequential datasets, especially the real-world ones, often don’t have the complete triplet data or with a lot of missing/noising data. So its generalizability should be further investigated.
>
> **A2:** Thank you for this remark. As we just mentioned, our proposed LaST does not require the trend and seasonal data as input. We reconstruct them from the representation $Z$ from the input time series $X$. Thus, most real-world datasets will meet the condition required by our model.  Besides, data missing problem is a critical challenge in the time series data modeling. We’d like to pay more attention of this problem in our future work.
>
> -------
>
> **Q3:**  The author might need to add more baselines (other disentangled representation learning methods for sequential data). Can you compare your results with other disentangled representation learning methods for sequential data?
>
> **A3:** Of course! We add a recently proposed sequential disentanglement representation learning model C-DSVAE [1] as a new baseline and compare it with our model. In fact, one of our existing baselines, CoST, is also a disentangled representation learning-based model which employs contrastive learning in frequency and temporal domain to obtain the disentangled representations. In the following two tables, we compare our model with C-DSVAE and CoST on ETTh1 and ETTm1 datasets:
>
> Table 1: Multivariate forecasting comparison with disentangled methods on ETTh1 dataset (metric: MSE/MAE).
>
> | Approach | 24 | 48 | 168 | 336 | 720 |
> | --- | --- | --- | --- | --- | --- |
> | C-DSVAE | 0.428/0.438 | 0.487/0.462 | 0.621/0.590 | 0.735/0.628 | 0.990/0.781 |
> | CoST | 0.389/0.429 | 0.437/0.464 | 0.643/0.582 | 0.812/0.679 | 0.812/0.679 |
> | LaST | 0.324/0.368 | 0.351/0.380 | 0.468/0.453 | 0.566/0.512 | 0.740/0.650 |
>
> Table 2: Multivariate forecasting comparison with disentangled methods on ETTm1 dataset (metric: MSE/MAE).
>
> | Approach | 24 | 48 | 96 | 288 | 672 |
> | --- | --- | --- | --- | --- | --- |
> | C-DSVAE | 0.429/0.423 | 0.576/0.510 | 0.584/0.513 | 0.600/0.535 | 0.618/0.547 |
> | CoST | 0.246/0.329 | 0.331/0.386 | 0.378/0.419 | 0.472/0.486 | 0.620/0.574 |
> | LaST | 0.218/0.289 | 0.280/0.329 | 0.323/0.360 | 0.392/0.403 | 0.491/0.466 |
>
> From the tables, we can observe LaST achieves the best performance in all forecasting horizons, demonstrating the effectiveness of our proposed disentanglement learning scheme. We will add the new results in the main text.
>
> [1] Junwen Bai, Weiran Wang, and Carla P. Gomes. Contrastively disentangled sequential variational autoencoder. NeurIPS, 2021, 10105-10118.

---

### Official Review · Reviewer_JNkP · 2022-07-15

**Rating:** 6
**Confidence:** 5
**Soundness:** 2 fair
**Presentation:** 3 good
**Contribution:** 2 fair

**Summary:**

This paper focuses on the time series forecasting task and presents a decomposition method LaST to learn latent representations. The decomposition method is derived from variational inference and mutual information. Along with a predictor, LaST can achieve competitive performance in many benchmarks.

**Questions:**

1. What is the difference between the autocorrelation calculation used in Equ 6 and the autocorrelation calculation method in Autoformer? Maybe giving a citation is better.
2. The visualization in Figure 2 is not surprising. You can conduct the same visualization to Autoformer. I think the simple moving average block can achieve the representation disentanglement well. You can compare these two decomposition methods.
3. I am a little confused about the predictor (Figure 1 (b)) without the supplementary material. Especially in lines 127-128, the inverse DFT cannot change the sequence length. You should give more details about the “extend” in the main text.
4. It seems the prediction results show the over-smoothing problem (Figure 3). This problem is fatal for details predicting of time series. This can be caused by the design of the predictor. More showcases are required especially the non-stationary time series.


**Ethics Review Area:**

["I don’t know"]

**Limitations:**

No, the author has not discussed any limitations of this work. I think the convergence property can be further analyzed. Also, the stochasticity of time series is neglected in the decomposition, which should also be noticed by the authors.

**Strengths And Weaknesses:**

### Strengths
1. The loss function for time series decomposition is reasonable and novel.

2. This paper is well-organized and clear.

3. The model performance is competitive and is with detailed analysis.

### Weaknesses
1. The ‘proof’ of the ‘Theorem’ 2 is too intuitive.
- The minimization between autocorrelations of $X$ and $\hat{X}^s$ is not convincing, which will bring the noise to the optimization. And this process also relies on an underlying assumption, that the raw time series is periodic by removing the trend, which is contradictory to the statement in line 89.
- The same problem is also in the design of CORT($X$,$\hat{X}^t$).
2. Comparing baselines:
- I think the N-BEATS is a necessary baseline since you both adopt the decomposition and a similar predictor for forecasting (linear model for trend and sine/cosine functions for seasonal).
- The most important thing of this paper is to explain the necessity of learning ‘latent’ representations. What about using the moving average of Autoformer for decomposition and adopting the same predictor for prediction? I think this vanilla baseline is also necessary.

---

> ### Author Response · Authors · 2022-08-02
> **Response to Reviewer JNkP -- Part 1**
>
> Many Thanks to Reviewer JNkP for the thorough review and valuable comments. The responses to your concerns are as below.
>
> --------------
>
> **Q1:** The 'proof' of the 'Theorem' 2 is too intuitive. (1) The minimization between autocorrelations of $\hat{X}$ and $\hat{X}^s$ is not convincing, which will bring the noise to the optimization. And this process also relies on an underlying assumption, that the raw time series is periodic by removing the trend, which is contradictory to the statement in line 89. (2) The same problem is also in the design of CORT.
>
> **A1**: Thanks for this great comment. We first respond to the concern about autocorrelations and CORT metrics. Starting from the design motivation, we require two evaluation standards for seasonal and trend characteristics as constraints that make two representations hold their corresponding semantics and further avoid entanglement. Thus, we choose autocorrelations to reflect the seasonal part. This metric denotes the similarity between every instant value in $X$ and the corresponding lagged value and is also used to extract seasonal features in many works [1], [2]. CORT focuses on the first differences between two input time series, which can further reflect their trend similarity. We agree that these two metrics are not strong enough to describe a time series thoroughly. In other words, we cannot determine a unique sequence convincingly only with autocorrelations and CORT. This is also the reason why we introduce an additional (MSE) loss, to ensure that the reconstruction is the same as the original signal. However, the purpose of using these two metrics is only to guide the optimization directions of seasonal and trend representations by constraining reconstructions $\hat{X}^t$ and $\hat{X}^s$. Ablation study results (cf. Table 3 in page 8) also support the observation that these two metrics can help enhance the forecasting performance. A formal treatment of the accidentally introduced noise is indeed an important aspect and they are interesting challenges for us to address in future work (e.g., how to detect and quantify the noises if any). we will add an explicit note to that effect in the final version, stating that we plan on investigating it.
>
> Regarding the statement in line 89, we believe that there is no contradiction. Similar to our proposed LaST, Autoformer also exploits the decomposition strategy for better time series forecasting. Our LaST and Autoformer both obey the same assumption that time series can be formed as the sum of seasonal and trend. The difference is that Autoformer decomposes seasonal and trend by a simple moving average pooling module while ours employ disentangled latent representations. Besides, LaST does not rely on a specific fixed window and thus is more flexible with input time series. Thanks again for your valuable comment. We will improve our description in the final version.
>
> [1] Michail Vlachos, Philip Yu, and Vittorio Castelli. On periodicity detection and structural periodic similarity. *SDM,* 2005, 449–460.
>
> [2] Michail Vlachos, Philip Yu, and Vittorio Castelli. A periodogram-based metric for time series classification. *Computational Statistics & Data Analysis*, 50:2668-2684, 2006.

---

> ### Author Response · Authors · 2022-08-02
> **Response to Reviewer JNkP -- Part 2**
>
> **Q2:** Comparing baselines: (1) I think the N-BEATS is a necessary baseline since you both adopt the decomposition and a similar predictor for forecasting (linear model for trend and sine/cosine functions for seasonal). (2) The most important thing of this paper is to explain the necessity of learning ‘latent’ representations. What about using the moving average of Autoformer for decomposition and adopting the same predictor for prediction? I think this vanilla baseline is also necessary.
>
> **A2**: Thanks for this great comment and we agree that these two baselines are necessary.
>
> N-BEATS is a univariate time series forecasting model. Thus, we provide comparisons between our model and N-BEATS on the univariate setting. The results are shown in the following two tables:
>
> Table 1: Univariate forecasting comparisons on ETTh1 dataset (metric: MSE/MAE).
>
> | Model | 24 | 48 | 168 | 336 | 720 |
> | --- | --- | --- | --- | --- | --- |
> | N-BEATS | 0.042/0.156 | 0.065/0.200 | 0.106/0.255 | 0.127/0.284 | 0.269/0.422 |
> | LaST | 0.030/0.131 | 0.051/0.169 | 0.078/0.211 | 0.100/0.246 | 0.138/0.298 |
>
> Table 2: Univariate forecasting comparisons on ETTm1 dataset (metric: MSE/MAE).
>
> | Model | 24 | 48 | 96 | 228 | 672 |
> | --- | --- | --- | --- | --- | --- |
> | N-BEATS | 0.031/0.117 | 0.056/0.168 | 0.095/0.234 | 0.157/0.311 | 0.207/0.370 |
> | LaST | 0.011/0.077 | 0.021/0.108 | 0.033/0.134 | 0.069/0.197 | 0.100/0.239 |
>
> The results indicate that LaST performs much better than N-BEATS on the univariate time series forecasting task.
>
> As per your request, we keep the predictor identical and further compare our decomposition approaches with that in Autoformer, i.e., moving average. The experimental results are as follows:
>
> Table 3: Multivariate forecasting comparison with different decomposition approaches on ETTh1 dataset (metric: MSE/MAE).
>
> | Approach | 24 | 48 | 168 | 336 | 720 |
> | --- | --- | --- | --- | --- | --- |
> | Moving Average | 0.364/0.398 | 0.384/0.406 | 0.556/0.516 | 0.690/0.598 | 0.976/0.777 |
> | Ours | 0.324/0.368 | 0.351/0.380 | 0.468/0.453 | 0.566/0.512 | 0.740/0.650 |
>
> Table 4: Multivariate forecasting comparison with different decomposition methods on ETTm1 dataset (metric: MSE/MAE).
>
> | Approach | 24 | 48 | 96 | 288 | 672 |
> | --- | --- | --- | --- | --- | --- |
> | Moving Average | 0.228/0.293 | 0.300/0.344 | 0.345/0.379 | 0.394/0.407 | 0.503/0.476 |
> | Ours | 0.218/0.289 | 0.280/0.329 | 0.323/0.360 | 0.392/0.403 | 0.491/0.466 |
>
> We can see that our approach outperforms Moving Average. We will include the above baselines and corresponding results in our main paper.
>
> ---------------
>
> **Q3:** What is the difference between the autocorrelation calculation used in Equ 6 and the autocorrelation calculation method in Autoformer? Maybe giving a citation is better.
>
> **A3:** Thanks for this suggestion. Our autocorrelation calculation, deriving from [1], is the same as that in Autoformer. Compared to direct calculation in temporal domain, it achieves the autocorrelation sequence in frequency domain and holds efficiency. The formulas for this calculation have been provided in Appendix B.2. We will add this citation in our final version.
>
> [1] George EP Box, Gwilym M. Jenkins, Gregory C. Reinsel, and Greta M. Ljung. Time series analysis: forecasting and control. John Wiley & Sons, 2015.
>
>
> ----------
>
> **Q4:** The visualization in Figure 2 is not surprising. You can conduct the same visualization to Autoformer. I think the simple moving average block can achieve the representation disentanglement well. You can compare these two decomposition methods.
>
> **A4:** Thanks for this good comment. We conducted this visualization to evaluate whether our mechanism can make seasonal-trend representations disentangled in latent space. Results (cf. figure 2) in the main paper support the success. As per your suggestion, we visualize the seasonal and trend representations in the decoder in supplementary materials (see “repre_visual.pdf”). The results show that their representations are muddled and cannot be distinguished well. This is because the decomposition mechanism of Autoformer paid attention to time series itself rather than representations. This visualization comparison suggests that: (1) learning disentangled seasonal-trend representations is not trivial, and (2) our proposed decomposition methods are effective. We will add these results and more discussions to the paper.

---

> ### Author Response · Authors · 2022-08-02
> **Response to Reviewer JNkP -- Part 3**
>
> **Q5:** I am a little confused about the predictor (Figure 1 (b)) without the supplementary material. Especially in lines 127-128, the inverse DFT cannot change the sequence length. You should give more details about the “extend” in the main text.
>
> **A5**. Sorry for the confusion. Our predictor is composed of seasonal and trend parts. The trend part consists of two linear transformations with two feed forward networks. It can be formulated by:
>
> $$
> \begin{align}
> \\tilde{Z}^t =FFN_1(Z^t),\\\\
> Y^t=FFN_2(\\tilde{Z}^t),
> \end{align}
> $$
>
> where dimensions transformations are $FFN_1:T\rightarrow \tau$ and $FFN_2: d_Z \rightarrow d$. Here $d_Z$ and $d$ denote dimensions of representations and time series, respectively. The seasonal part first exploits the discrete Fourier transform (DFT) and inverse DFT (iDFT), which, as Appendix B.1 shows, can be formulated by:
>
>
>
> $$
> \begin{align}
> &Z_{\mathcal{F},k}^s  = \mathcal{F}(Z^s)_k = \sum _{t=1}^T \cdot \exp(\frac{-2\pi i k t}{T}),\qquad 1\leq k\leq \left \lfloor \frac{T+1}{2} \right \rfloor \\\\
> & Z _{t}^s = \mathcal{F}^{-1}(Z _{\mathcal{F}^s}) _{t} = \frac{1}{T} \sum _{k=1}^{T}Z _{\mathcal{F},k}^s\cdot \exp(\frac{2\pi i kt}{T}),  \qquad 1 \leq t \leq T + \tau.
> \end{align}
> $$
>
> With these two formulas, sequence length has been extended, and thus we can derive predictable representations $\tilde{Z}^s$ by taking the last $\tau$ time steps. Then we transform it into seasonal forecastings:
>
> $$
> Y^s=FFN(\tilde{Z}^s),
> $$
>
> where, same as the trend predictor, $FFN: d_Z \rightarrow d$. Finally, we obtain the forecasting by adding these two parts $Y=Y^s+Y^t$. We will provide more details in the main text. Thank you for this great suggestion!
>
> ------------------
>
> **Q6:**  It seems the prediction results show the over-smoothing problem (Figure 3). This problem is fatal for details predicting of time series. This can be caused by the design of the predictor. More showcases are required especially the non-stationary time series.
>
> **A6**. Thanks for this insightful comment. As per your request, we have added some non-stationary time series in supplementary materials (see “non_stationary_cases.pdf”) to further validate the capability of our model. It shows that facing non-stationary time series, the reconstructed seasonal and trend still jointly restore the time series with their own characteristics. We’d be happy to extend these cases in our final version.

---

> ### Comment · Reviewer_JNkP · 2022-08-03
> **Thanks for your response**
>
> Most of my concerns are addressed by the authors. I still have two concerns.
>
> - As stated in the original review, I also have questions about the "limitations". And I cannot find any discussions about this in the revised paper.
> - About the future work, maybe imputation shares some similarities with forecasting, but anomaly detection is totally a different area w.r.t. forecasting (Classification v.s. Regression). You may change this description.
>
> In general, I appreciate this paper's contribution to time series forecasting. I am willing to raise the score if you address the above two questions.

---

> > ### Author Response · Authors · 2022-08-05
> > **Author Response to Remaining Concerns**
> >
> > Thanks for your response and appreciation. Here are the responses to your remaining concerns.
> >
> > ---
> >
> > **Q1:** I think the convergence property can be further analyzed.
> >
> > **A1:** Thanks for the good suggestions. As per your request, we have conducted additional experiments to validate the convergence of our model and put the results into our supplementary materials (see “convergence.pdf”). Figure 1 shows the descending process of training, valid, and test loss of our model as the epochs increase, where we can see that all losses first drop and then goes to level off. The downward trend is not obvious on ETTm1 dataset. This is because ETTm1 is a larger-scale dataset and it requires over 1,000 times stochastic gradient descent at every epoch. We visualize the losses change at the first epoch in figure 2. We can observe that losses drop a lot at the first epoch. These results demonstrate that our model converges well on real-world datasets. We will add these analyses in our final version.
> >
> > ---
> >
> > **Q2:** The stochasticity of time series is neglected in the decomposition, which should also be noticed by the authors.
> >
> > **A2:** Thanks for the comment. In our paper, we have a simple assumption that time series can be decomposed into seasonal and trend parts. It is a classical decomposition strategy and achieves success in time series forecasting. However, as your mentioned, it does ignore the stochasticity which usually appears in real-world datasets. On the one hand, variational inference implicitly consists of Gaussian noise when establishing normal distribution, which simulates stochasticity to some extent, but the effect is limited. On the other hand, considering the stochasticity is also a challenging work in time series. Some outstanding papers such as [1] have made efforts to model it. In our future work, we may consider modeling stochasticity explicitly. We will clarify this point in our paper. Thanks for your suggestions again!
> >
> > [1] Sun, Fan-Keng, Chris Lang, and Duane Boning. "Adjusting for autocorrelated errors in neural networks for time series." *Advances in Neural Information Processing Systems*
> >  34 (2021): 29806-29819.
> >
> > ---
> >
> > **Q3**: About the future work, maybe imputation shares some similarities with forecasting, but anomaly detection is totally a different area w.r.t. forecasting (Classification v.s. Regression). You may change this description.
> >
> > **A3:** Thanks for the helpful suggestions. Except for dealing with the time series, designing a good score function or classifier is of vital importance for an advanced anomaly detection model. We will point out the similarities and differences between these two tasks and organize the description carefully.

---

> > > ### Comment · Reviewer_JNkP · 2022-08-06
> > > **Thanks for your further response.**
> > >
> > > Thanks for your further response. I have raised my score from 5 to 6.

---

### Official Review · Reviewer_r48g · 2022-07-16

**Rating:** 6
**Confidence:** 4
**Soundness:** 3 good
**Presentation:** 4 excellent
**Contribution:** 3 good

**Summary:**

In this submission, the authors propose to learn disentangled seasonal and trend representations of seasonal time series. In particular, in contrast to previous methods that use average pooling for the trend feature, the major contribution here is to directly optimize the two representations by variational method. Also, a detailed analysis of the loss function and the optimization bound is provided. Empirical results show strong effectiveness and outperforms several strong baselines. The following analysis also supports the motivation.

**Questions:**

1. When discussing the connection to previous papers, the writing should both include the difference and the inheritance. Which components are extended from the previous studies? Also, what are techniques that can potentially be combined (such as contrastive learning from CoST)?

2. What is the boundary of the method? The drawback of feature degeneration should be discussed in more detail.

3. What is the training and inference time? How it can be compared with the baselines?

**Limitations:**

As discussed above, the reviewer encourages the authors to face the general limitation of VI methods and discuss the implications in this setting.

**Strengths And Weaknesses:**

Originality:
The proposed method of representation disentanglement is an extension of other fields such as computer vision. However, given the nature of seasonal time series, the application of this method is non-trivial, as well as the design of a proper predictor. The bound proof is original in this setting.

Quality:
1. The conduct of empirical analysis is of high quality. The experiments are sufficient to support the claims (great performance boost). One weakness, as also noticed by the author, is the potential degeneration of features caused by narrowing down prior and posterior, and the effect is not discussed in detail.
2. As the reviewer is not an expert in theoretical analysis, I would leave this part to other reviewers.

Clarity:
1. The overall presentation is clear, including a good schematic pipeline description and results tables.
2. The development of Theorem 2 is a bit confusing. The authors first claim that L rec can be estimated without leveraging Xs and Xt, while eq 6 still has the two variables. Line 143, what is "first difference?"

Significance:
The reviewer thinks this method provides significant advances in this subfield.

---

> ### Author Response · Authors · 2022-08-02
> **Response to Reviewer r48g -- Part 1**
>
> Many thanks to Reviewer r48g for the thorough review and valuable comments. The responses to your concerns are as below.
>
> -------
>
> **Q1:** One weakness, as also noticed by the author, is the potential degeneration of features caused by narrowing down prior and posterior, and the effect is not discussed in detail. The drawback of feature degeneration should be discussed in more detail.
>
> **A1:** Thanks for the insightful comment. We agree that potential feature degeneration exists when narrowing down the prior and posterior. This is because the widely used variational inference [1] maximizes the likelihood $P(X,Y)$ with evidence lower bound (ELBO) term, which minimizes the KL divergence between the prior $P(Z)$ and $Q(Z|X)$. Theoretically, the $Q(Z|X)$ will be non-informative for the inputs $X$ if the capacity of neural networks is strong enough. This situation leads to useless forecastings for the input time series. However, we cannot simply remove this term: we should ensure it is still an effective lower bound.
>
> Thus, to tackle this feature degeneration problem, we introduced the mutual information (MI) between observation $X$ and latent representation $Z$ and try to maximize this term to maintain the correlations between $X$ and $Z$. The detailed maximization method and technique are shown in Sect. 4.
>
> Besides, the experimental results of “*w/o lower bound”* in ablation study (cf. Table 3 in page 8) can be seen as negative effects of feature degeneration. Without the mutual information constraint, the performance drops ~4% on average, and in some metrics (e.g., MSE with horizon 168 on ETTh1 dataset), this variant drops ~10%. This result, in a sense, additionally supports the claim of effectiveness of our solution. We will provide more detailed discussions of feature degeneration.
>
> [1] Diederik P. Kingma and Max Welling. Auto-encoding Variational Bayes. ICLR, 2014.
>
> -------
>
> **Q2:**  The development of Theorem 2 is a bit confusing. The authors first claim that L rec can be estimated without leveraging $X^s$ and $X^t$, while eq 6 still has the two variables. Line 143, what is “first difference?”
>
> **A2:** Thank you for pointing this out, and we wish to apologize for the confusion. Since the seasonal $X^S$ and trend $X^t$ cannot be obtained as input, we use $\hat{X}^s$ and $\hat{X}^t$ to replace them in Eq. (6) and in Line 143, where the $\hat{\cdot}$ mark denotes the corresponding reconstructions from the variational inference. We will discriminate between these two marks more clearly in the final version. Besides, $\Delta X^t$ should be $\Delta X$ in Eq. (7). Thanks for the careful review.
>
> The first difference denotes the variation during the time series. In a time series $X_{1:T}$ with length $T$, it is also a series and can be obtained by $\\{X_{t+1}-X_{t} \\} _{t\in [1:T-1]}$. Clearly, this reflects the trends of time series, and we will provide more detailed explanation in the final version.

---

> ### Author Response · Authors · 2022-08-02
> **Response to Reviewer r48g -- Part 2**
>
> **Q3:** When discussing the connection to previous papers, the writing should both include the difference and the inheritance. Which components are extended from the previous studies? Also, what are techniques that can potentially be combined (such as contrastive learning from CoST)?
>
> **A3:** Thanks for the helpful comment. As the reviewer has pointed out, CoST exploits the contrastive learning method to encourage the discriminative of seasonal and trend representations. Autoformer employs a moving average pooling module to decompose time series directly. Different from these two approaches, LaST designs two seasonal and trend metrics and minimize the mutual information to further disentangle the seasonal and trend representations.
>
> Besides, we are inspired by CoST to design the seasonal predictor with the discrete Fourier transform (DFT) mechanism. Autoformer and our LaST both use Autocorrelation to reflect the seasonal patterns of time series. However, Autoformer designs it as a kind of score to establish the correlations between time steps, while LaST employs it to measure the seasonal reconstruction. We will add these comparisons in our final version to assist readers better understand the background and connections of our model with previous approaches. Thank you for this great suggestion.
>
> -------
>
> **Q4:**  What is the training and inference time? How it can be compared with the baselines?
>
> **A4:** Thanks for this comment. We have conducted experiments to compare the running time with advanced time series models. We keep the batch size identical and record the time consumption of an epoch. The results are concluded as follows:
>
> Table 1: Training and inference time consumption comparisons on ETTh1 datasets in different forecasting horizons.
>
> | Method | 24 | 168 | 720 |
> | --- | --- | --- | --- |
> | CoST | 8.23 s | 9.40 s | 13.13 s |
> | LaST | 14.88 s | 18.75 s | 24.53 s |
> | Autoformer | 19.41 s | 35.44 s | 83.57 s |
>
> As the table shows, Autoformer needs more running time especially in large horizons, since the complexity of its attention mechanism is squared with forecasting horizons, while the complexities of LaST and CoST are linear with that. Besides, the training process of our model on ETTh1 dataset only requires about 20 epochs to converge. Thus, the complete training at a specific forecasting horizon only takes 10-minutes of consumption. We will add these results to supplementary materials in our final version. Thanks for your suggestions again!

---

### Official Review · Reviewer_hjLn · 2022-07-18

**Rating:** 7
**Confidence:** 5
**Soundness:** 3 good
**Presentation:** 4 excellent
**Contribution:** 3 good

**Summary:**

The paper presents an approach for learning latent season-trend based representations for time-series forecasting, based on variational inference. A sesonal encoder and a trend encoder are used to obtain latent representations. The latent representations are then combined together to produce a forecast. The forecast is done via the predictor module, which makes use of Fourier Transform for the seaosnal representation and a simple MLP for a the trend representation, and then sums them up to obtain the forecast. Reconstruction error cannot be minimized directly, as the true trend and season components are not known, as a result autocorrelation distance is used for seaosnality reconstruction, and the temporal correlation is used for reconstructing the trend reconstruction. For the purpose of disentanglement, the MI between the latent components is minimized. The overall loss function has an ELBO loss, and MI maximization between the original time-series and latent component, and MI minimization between the season and trend latent components. Since the optimization is untractable, lower and upper bounds for MI are derived to help with the optimization. Experiments are performed on standard univariate and multivariate timeseries benchmarks, showing LaST achieves state of the art performance. Visualizations show effective disentanglement.

**Questions:**

- how complex was performing this training and getting it to work? To me, it seemed like a daunting task

**Strengths And Weaknesses:**

Strengths
- good motivation to expand on decomposition of trend seasonality
- novel way to do so compared to existing approaches (e.g. variational inference vs CoST using contrastive learning)
- meaningful loss function (elbo, reconstruction, predictor)
- identified technical challenges in reconstruction, and proposed a way to address it
- identified challenges in MI optimisation and developed bounds to address it
- good coverage and comprehensive experiments - baselines + datasets
- good ablation / visualizations
- a generally comprehensive paper which is well written

Weakness
- superposition of a lot of complex components to make it work, possibly creating uncertainty on the robustness of the model

---

> ### Author Response · Authors · 2022-08-02
> **Response to Reviewer hjLn**
>
> Many thanks to Reviewer hjLn for the thorough review and valuable comments. The responses to your concerns are as below.
>
> -------
>
> **Q1:** Superposition of a lot of complex components to make it work, possibly creating uncertainty on the robustness of the model.
>
> **A1:** Thank you for raising this concern. We agree that in our model, there are some complex components like the Fourier transform and its inverse transform in our predictor. We would like to mention that most of the improvements we proposed, including the autocorrelation and mutual information terms, are only involved in the training phase as optimization targets. Once the model is trained, these components are not included in the time series forecasting phase. We also provide ablation analysis for each component in Table 3 in the main context to validate their improvement.
>
> For the robustness of our model, we have provided the standard deviations in Table F3 (Fluctuation analysis) in Appendix F. We run baselines and our model three times on all datasets to observe the performance fluctuations. The table shows that performance changes are within 3% for most datasets. Even on the small-size dataset “exchange rate”, the error records are also within 10%. These fluctuation ranges are much smaller than the enhancement our model achieves. Thus we believe our model is as robust as baselines.
>
> -------
>
> **Q2:** How complex was performing this training and getting it to work? To me, it seemed like a daunting task.
>
> **A2:** Thanks for this comment. The computational complexity is actually a very important property for evaluating the overall quality of the model. We have analyzed the complexity of the training process in Appendix E.1. We have conducted additional experiments and measured the training time and the number of epochs that the model needs for converging. The results are shown in Table 1 below, where we can see that our proposed LaST is efficient and can reach convergence within 10 minutes. We’d be happy to add it to the supplementary materials and indicate it in the main text.
>
> Table 1: Training and inference time consumption and the number of epochs on the ETTh1 dataset in different forecasting horizons.
>
> | Horizon | 24 | 168 | 720 |
> | --- | --- | --- | --- |
> | Epoch time | 14.88 s | 18.75 s | 24.53 s |
> | # Epochs | 26 | 16 | 17 |

---

### Comment · Area_Chair_c9zG · 2022-08-09
**Please acknowledge the authors’ rebuttal and update your review if necessary**

Dear Reviewers,

The authors have provided the rebuttal responses. The discussion period between authors and reviewers will end soon.
Please do check the author's response, acknowledge your reading, and update your review if needed.
If there is any further question, please do ask the authors to clarify before the discussion period ends.

Thank you for your professional service!

Your AC

---

### Meta-Review · Area_Chair_c9zG · 2022-08-26

**Recommendation:** Accept
**Confidence:** Certain

**Metareview:**

The paper presents a novel learning approach named LaST for time-series forecasting based on variational inference to disentangle the seasonal-trend representations in the latent space. Empirical results validate the effectiveness of the proposed method in comparison with several strong baselines. Reviewers generally agree the work is technically solid, the idea is novel, the experiments are convincing, and the paper is well presented. Thus, the paper is clearly above the acceptance bar. Authors are encouraged to incorporate all the discussions and the additional results during the rebuttal in the final version.

**Award:**

No

---

### Decision · Program_Chairs · 2022-09-14

Accept